



# 3D Wind Vector Measurements using a 5-hole Probe with Remotely Piloted Aircraft

Radiance Calmer[1], Greg Roberts[1,2], Jana Preissler[3], Solène Derrien[4], and Colin O'Dowd[3]

[1]CNRM UMR, Météo-France/CNRS,Toulouse, France;
[2]Scripps Institution of Oceanography, University of California, San Diego, CA;
[3]School of Physics and Centre for Climate and Air Pollution Studies, National University of Ireland Galway, Ireland;
[4]Laboratoire d'Aerologie, University of Toulouse, CNRS, France

*Correspondence to:* Radiance Calmer (radiance.calmer@meteo.fr)

Submission to Atmospheric Measurement Techniques journal

**Abstract.** The importance of 3D winds (in particular updraft) in atmospheric science has motivated the adaptation of airborne wind instruments developed for manned aircraft, to the small size of Remotely Piloted Aircraft Systems (RPAS).

5   Simultaneously, enhancements in RPAS technology have increased their contribution to many fields. In atmospheric research, lightweight RPAS ($< 2.5$ kg) are now able to accurately measure 3D wind vectors, even in a cloud, which provides new observing tools for understanding aerosol-cloud interactions. The European project BACCHUS (Impact of Biogenic versus Anthropogenic Emissions on Clouds and Climate: towards a Holistic Understanding) focuses on these specific interactions. Vertical wind velocity at cloud base is a key parameter for aerosol-cloud interactions. To measure the three components of

10   wind, one RPAS is equipped with a 5-hole probe and an Inertial Measurement Unit (IMU), synchronized on an acquisition system. The 5-hole probe is calibrated and validated on a multi-axis platform in a wind tunnel, each probe and its associated pressure sensors have specific calibration coefficients. Once mounted on a RPAS, 3D winds and turbulent kinetic energy (TKE) derived from the 5-hole probe are validated with a sonic anemometer on a meteorological mast. During the BACCHUS field campaign at Mace Head (Ireland), a fleet of RPAS has been utilized to profile the atmosphere and complement

15   ground-based and satellite observations. To study aerosol-cloud interactions, the RPAS with the 5-hole probe flew at level legs near cloud base to measure vertical wind speeds. The vertical velocity measurements from RPAS are validated with vertical velocities derived from the Mace Head Doppler cloud radar, and the results illustrate the relationships between the distributions of vertical velocity and the different cloud fields.



# 1  Introduction

Three dimensional wind vectors are an essential parameter for understanding atmospheric processes such as aerosol-cloud interactions and boundary layer turbulence. In tracing the evolution of aircraft-based wind measurements in the atmosphere, three axes of development have been pursued since the 1960s: airborne platforms, inertial navigation systems (INS) and sensors. Airborne platforms have evolved from large aircraft (i.e., Canberra PR3; Axford (1968) or NCAR Queen Air; Brown et al. (1983)) to ultra-light unmanned aerial systems (i.e., SUMO; Reuder et al. (2008)). INS measure six axes of aircraft motion, and are used to back out wind vectors in the Earth's coordinate frame. A major improvement in INS was the integration of GPS data with fusion sensors (Khelif et al., 1999). The overall accuracy of 3D wind vectors has improved drastically, from 1 m s$^{-1}$ with wind vanes (Lenschow and Spyers-Duran, 1989) to 0.03 m s$^{-1}$ with a multi-hole probe (Garman et al., 2006). Since the past decade, GPS, INS and meteorological sensors have become sufficiently miniaturized to be deployed on ultra-light remotely piloted aircraft systems (RPAS)[1], which has extended observational capabilities previously limited to traditional manned aircraft.

A wide range of RPAS has been used to measure atmospheric winds, from a 30 kg Manta (Thomas et al., 2012) to a 600 g SUMO (Reuder et al., 2008). In particular, a multi-hole probe paired with an inertial measurement unit (IMU; equivalent to INS with integrated GPS data) has been the main mechanism for obtaining 3D winds in fixed-wing RPAS. Ultimately, the combination of multi-hole probe, differential pressure measurements and IMU dictates the precision of atmospheric wind measurements. A lab-built 5-hole probe provides a measurement of vertical velocity within 0.4 m s$^{-1}$ combined with a GPS-MEMS IMU (van den Kroonenberg et al. (2008); M$^2$AV-Spiess et al. (2007); MASC-Wildmann et al. (2014)), and 0.5 m s$^{-1}$ with a SBG System IMU (ALADINA-Altstädter et al. (2015)). The Aeroprobe, a commercially available 5-hole probe implemented on a Manta platform with a C-Migits-III tactical sensor as IMU, obtained a measurement of vertical velocity to within 0.17 m s$^{-1}$ on vertical wind (Thomas et al., 2012). A 5-hole Aeroprobe was also used on SUMO RPAS combined with the IMU embedded in the autopilot navigation system to retrieve wind. The uncertainty on 3D wind measurement was provided for the probe reference frame as 0.1 m s$^{-1}$ (Båserud et al., 2014). A fully-integrated 9-hole probe (with pressure sensors embedded in the probe) has been operated on Manta and ScanEagle RPAS with NovAtel IMU with relatively high precision of 0.021 m s$^{-1}$ for vertical wind speed and 0.045 m s$^{-1}$ for horizontal wind (Reineman et al., 2013).

Elston et al. (2015) has identified five main points that still need to be addressed for 3D wind measurements using RPAS: (1) true heading remains one of the main sources of inaccuracy in horizontal wind calculation; (2) precise altitude with GPS; (3) miniaturization of IMU for small RPAS, with better accuracy of fusion sensors; (4) improved algorithms for wind field estimation from dynamic soaring; and (5) RPAS regulations and integration in the airspace, which can delay research progress.

---

[1]Commonly called unmanned aerial vehicle (UAV)



Until recently, wind measurements from RPAS have been mainly used for the atmospheric boundary layer to study turbulence and atmospheric fluxes. In the BLLAST field campaign, multiple RPAS have been deployed to study the boundary layer during the transition between afternoon and evening periods (Lothon et al., 2014). Results of sensible and latent heat fluxes, and also turbulent kinetic energy (TKE), were calculated from the SUMO RPAS flights, (Reuder et al. (2016); Båserud

et al. (2016)). Operation of the $M^2AV$ and the MASC RPAS during the BLLAST campaign was described in Lampert et al. (2016) with a particular focus on turbulence. TKE decreased along the afternoon-evening transition to reach a minimum near sunset, and turbulence isotropy depended on the presence of a low-level jet. A comparison of near co-located measurements of TKE between different platforms (tethered balloon, RPAS, and manned aircraft) validated the different techniques of obtaining 3D wind vectors (Canut et al., 2016).

In addition to the boundary layer studies, aerosol-cloud interactions (ACI) remain one of the main uncertainties in understanding atmospheric processes (Boucher et al., 2013), which is the focus of the collaborative project, BACCHUS (impact of Biogenic versus Anthropogenic emissions on Clouds and Climate : towards a Holistic UnderStanding) (BACCHUS, 2016). The study presented hereafter is part of the BACCHUS project, and presents results from a fleet of RPAS

instrumented to study aerosol-cloud interactions. One critical parameter in ACI studies, not previously measured by RPAS, is the vertical velocity $w$ at cloud base, which has been identified as essential to quantifying the aerosol effect on cloud properties (Hudson and Svensson (1995); Snider and Brenguier (2000); Schmidt et al. (2015)). Peng et al. (2005) showed the importance of vertical velocity for convective clouds in a cloud closure study, and highlighted the need of more cloud microphysical data to further test the sensitivity of cloud droplet number concentration to variations in vertical velocity. In

Conant et al. (2004) and Sanchez et al. (2017), updraft has also been described as a critical parameter, along with cloud condensation nuclei (CCN) spectra, to derive cloud droplet number concentration (CDNC) in ACI studies.

The motivation of this present work is to assess the ability of RPAS to measure vertical wind velocity near cloud base to study aerosol-cloud interactions. The first sections of the manuscript describe the RPAS platform and the methods used to

calculate 3D winds. Then, the details of calibration of the 5-hole probe in a wind tunnel are discussed, complemented by an uncertainty analysis on vertical wind velocity, $w$ (the main parameter needed for ACI studies). A comparison with a sonic anemometer on a meteorological mast provides a validation of RPAS measurements in relatively calm wind conditions. Lastly, several case studies focus on flights under a range of turbulent conditions, during a BACCHUS field campaign in Ireland, and vertical wind velocities from the RPAS are compared to those of a cloud radar.



## 2 Materials and methods

### 2.1 Remotely Piloted Aircraft System (RPAS) description

The RPAS used here to measure 3D winds and study aerosol-cloud interactions are based on the commercially available Skywalker X6 model. The wingspan is 1.5 m long, and take-off weight varies between 1.5 kg and 2.3 kg depending on the

mission specific payload. The navigation system is the open source autopilot Paparazzi from Ecole Nationale de l'Aviation Civile (Brisset et al., 2006). One of RPAS (wind-RPAS) is specially equipped to measure three dimensional wind vectors, whose validation and study of different cloud cases is the purpose of this work. Its take-off weight is 1.5 kg for a payload of 500 g with cruise airspeed approximately 16 m s$^{-1}$ .

### 2.2 Instrumentation

The payload of the wind-RPAS to measure 3D wind vector is composed of temperature (IST, Model P1K0.161.6W.Y.010), pressure (All Sensors, Model 15PSI-A-HGRADE-SMINI) and relative humidity sensors (IST, P14 Rapid-W). Two Licor LI-200R pyranometers are installed on the fuselage; one facing up to measure downwelling solar irradiance, and the other facing down to measure upwelling solar irradiance, and are used to detect presence of cloud. Wind vectors are obtained from a 5-hole probe linked to its pressure sensors (All Sensors) by tubing, and an Inertial Measurement Unit, IMU (Lord Sensing Microstrain

3DM-GX4-45). The data from both the IMU and the pressure sensors are recorded by the same acquisition system to ensure precise synchronization. The acquisition frequency is 30 Hz, and data are averaged to 10 Hz for analysis. The 5-hole probe is constructed by the Aeroprobe Corporation, and consists of stainless tube with a semi-spherical tip (Fig.1). The associated electronics have been designed at the Centre National de Recherches Météorologiques (CNRM) laboratory, and consist of three differential pressure sensors (All-Sensors 5inch-D1-MV) and one absolute pressure sensor (All Sensors MLV-015A). Figure

1a illustrates the probe schematic: hole 1 measures the total pressure; the differential pressure between holes 2 and 3 provides $\beta$, the angle of sideslip; the differential pressure between 4 and 5 gives $\alpha$, the angle of attack; and hole 6, a ring around the probe, corresponds to the static pressure port. The difference between total pressure (hole 1) and static pressure (hole 6) gives the dynamic pressure, and determines the airspeed, $V_a$. To obtain angles in degree and airspeed in m s$^{-1}$, the 5-hole probe system must be calibrated in the probe's coordinate system and converted to the Earth's coordinate system. The IMU sends

information to the acquisition system regarding attitude angles, roll $\phi$, yaw $\psi$ and pitch $\theta$, GPS time and GPS position and altitude, and ground speed of the RPAS in Earth's coordinate system. Schematics of coordinate systems and angles are shown in Fig.1b and also described in Lenschow and Spyers-Duran (1989), Boiffier (1998) or van den Kroonenberg et al. (2008).

### 2.3 Methods

3D wind vector in the Earth's coordinate system is obtained by subtracting the measured motion of the plane (given by the

IMU), from the motion of the air (given by the 5-hole probe). The measurement of 3D winds involves the fusion of three coordinate systems. The angle of attack $\alpha$, the angle of sideslip $\beta$, and the airspeed $V_a$ are measured by the 5-hole probe





in the probe coordinate system; while the attitude angles $\theta$, $\psi$ and $\phi$ are given in the RPAS coordinate system. Lenschow equations (Lenschow and Spyers-Duran, 1989) are then used to calculate the wind vector in the Earth's coordinate system. The angular acceleration of the RPAS is negligible, because the distance between the 5-hole probe and the IMU is on the order of centimeters. In addition, we only consider data from straight and level flight for the study here, which simplifies the Lenschow
equations to:

$$u = -V_a \times \sin(\psi + \beta) + V_e \qquad (1a)$$
$$v = -V_a \times \cos(\psi + \beta) + V_n \qquad (1b)$$
$$w = -V_a \times \sin(\theta - \alpha) + V_p \qquad (1c)$$

The 3D wind vectors $u$, $v$, $w$ are given in the Earth's coordinate system in Eq.(1), and are the three components of the
wind on x, y and z-axis, respectively (Fig.1b). The positive x-axis represents East, the positive y-axis represents North, and the positive z-axis represents upward direction. $V_e$, $V_n$, $V_p$ are provided by IMU, and are the RPAS ground velocities along the x, y and z-axis. $\psi$ and $\theta$ are the yaw and the pitch angle, respectively, determined by the IMU. $\alpha$ is the angle of attack, $\beta$ is the angle of sideslip, and $V_a$ represents the airspeed, provided by the 5-hole probe, and initially measured in the 5-hole probe coordinate system. Equation (1) takes into account rotations between the different coordinate systems to eventually provide 3D
wind vectors in the Earth's coordinate system.

## 3   Calibration of the 5-hole probe

The calibration of the 5-hole probe consists of a series of wind tunnel experiments, a wind tunnel (Theodor Friedrichs & Co) with a cross section of 70 cm, and a wind range between 0.15 to 50 m s$^{-1}$. The uncertainty associated to the wind velocity in the wind tunnel is less than 2 %. The calibration of the 5-hole probe is a two-step process — first establishing the relationship
between the absolute pressure sensor, raw voltage to mbar with a barometer, then associating the differential pressure in mbar to angles ($\alpha$ and $\beta$) or velocity ($V_a$) in degree or m s$^{-1}$, respectively. The probe, its electronics, and the IMU are installed on a two-axis platform with motion in vertical and horizontal planes. The multi-axis platform rotates in the pitch axis (motion in the vertical plane) and yaw axis (motion in the horizontal plane), controlled with a LabView program (Fig.2). The amplitude of pitch and yaw angles varies from -15 deg to 15 deg to simulate flight conditions.

### 3.1   Static calibration

The angle of attack $\alpha$ and the angle of sideslip $\beta$ are obtained from linear relationship between IMU angles and ratios of differential pressure sensors. The ratios are defined by the following relationships:





$$C_\alpha = \frac{\Delta(P_4 - P_5)}{\Delta(P_1 - P_6)} \quad \text{and} \quad C_\beta = \frac{\Delta(P_2 - P_3)}{\Delta(P_1 - P_6)} \tag{2}$$

$\Delta(P_2 - P_3)$ is the differential pressure between holes 2 and 3, related to the calculation of the angle of sideslip $\beta$; $\Delta(P_4 - P_5)$ is the differential pressure between holes 4 and 5, related to the calculation of the angle of attack $\alpha$; and $\Delta(P_1 - P_6)$ is the differential pressure between holes 1 and 6, related to the airspeed. The linear relationship between $C_\alpha$ and $C_\beta$ (5-hole probe) and the yaw angle $\psi$ and the pitch angle $\theta$ (IMU) determines the calibration coefficients (Fig.3).

$$C\alpha_1 = m_1 \times (\alpha + \alpha_0) + n_1 \quad \text{and} \quad C\alpha_2 = m_2 \times (-\alpha + \alpha_0) + n_2 \tag{3a}$$

$$C\beta_1 = k_1 \times (\beta + \beta_0) + j_1 \quad \text{and} \quad C\beta_2 = k_2 \times (-\beta + \beta_0) + j_2 \tag{3b}$$

$$\alpha_0 = \frac{n_1 - n_2}{m_1 - m_2} \quad \text{and} \quad \beta_0 = \frac{j_1 - j_2}{k_1 - k_2} \tag{3c}$$

The linear calibration coefficients are denoted by $m$, $n$, $j$, $k$ while $\alpha_0$ and $\beta_0$ are offsets in $\alpha$ and $\beta$ associated to the alignment of the pressure ports on the probe. In the calibrations performed here (Fig.3), $\alpha_0$ and $\beta_0$ are found to be -0.76 deg and -0.84 deg, respectively. Subscript 1 corresponds to measurements with the probe in its standard orientation, while subscript 2 corresponds to measurements with the probe in its inverted orientation (i.e., rotated 180 deg). To calculate the calibration coefficients $m$ and $n$ for the angle of attack, the yaw angle was held constant to zero while the pitch angle varied, and vice versa to obtain the calibration coefficients $k$, and $j$ for the angle of sideslip. In the wind tunnel, the $\alpha$ (5-hole probe) and the $\theta$ (IMU) angles are the same, as for $\beta$ (5-hole probe) and $\psi$ (IMU) angles. To account for offsets in the alignment, experiments are performed with the probe in the standard orientation shown in Fig.1, with roll angle equal to 0 deg (subscript 1), and the probe in inverted orientation for the roll angle equal to 180 deg (subscript 2). Likewise, the same procedure is followed with 90 deg and -90 deg to determine $\beta_0$. IMU angles and ratios of differential pressure sensors are recorded for platform positions between -15 deg and 15 deg for three air speeds in the wind tunnel. Figure 3 shows that calibration coefficients do not change for the range of airspeeds in these experiments (between 15 and 25 m s⁻¹).

In an experiment to verify linearity between the calibration coefficients ($C_\alpha$, $C_\beta$) and the range of angles ($\alpha$, $\beta$) supported for flights, the pitch and yaw angles of the multi-axis platform (Fig.4) are varied concurrently. Figure 4a shows the applied values of IMU pitch and yaw angles to the platform, and Fig.4b illustrates the corresponding values of $C_\alpha$, $C_\beta$ from the 5-hole probe. The experiment is conducted at a constant wind speed of 15 m s⁻¹. The slight angular bias in Fig.4 is due to a 3 degree roll angle in the mounting of the two-axis platform in the wind tunnel. In Fig.4b, non-linearity is observed when $C_\alpha$ or $C_\beta$ exceed $\pm1$, which corresponds to $\alpha$ and $\beta$ angles between $\pm10$ deg. Thus, 5-hole probe accurately measures wind vectors during RPAS flights when $\alpha$ and $\beta$ are within $\pm10$ deg; consequently, values of $\alpha$ or $\beta$ exceeding $\pm10$ deg are not used for final wind calculation.





The calculation of airspeed from the 5-hole probe, Eq.(4a to 4c), with speed of sound $a$ and Mach number $M$ (Anderson, 2001) has been proposed by the Aeroprobe Corporation.

$$V_a = a \times M \tag{4a}$$

$$a = \sqrt{\gamma R T} \tag{4b}$$

$$M = \sqrt{\frac{2}{\gamma - 1}\left(\left(\frac{P_t}{P_s}\right)^{\frac{(\gamma-1)}{\gamma}} - 1\right)} \tag{4c}$$

Where $\gamma = 1.4$ is the specific heat ratio, $R = 287.07$ J kg$^{-1}$ K$^{-1}$, $T$ is the temperature in Kelvin, $P_t$ is the total pressure in mbar, and $P_s$ is the static pressure in mbar. The absolute pressure at the tip of the 5-hole probe (hole 1; Fig.1a) is $P_t = \Delta(P_1 - P_6) + P_6$. The static pressure is measured at hole 6 with $P_s = P_6$. The probe airspeed was calibrated using the wind tunnel between 12 and 34 m s$^{-1}$ velocities. Equations (4) account for temperature $T$ and absolute total pressure $P_t$, and are

within 7 % of nominal value. The airspeed calculated with the Bernoulli's law yielded similar results within 10 %.

## 3.2 Dynamic calibration

After performing the static calibration (Section 3.1; Fig.3 and Fig.4), the validation of the 5-hole probe is conducted in the wind tunnel, imposing a triangular motion of the pitch angle with different amplitudes and frequencies (Fig.5, inset). The wind tunnel provides a laminar flow, thus updrafts and downdrafts are, by definition, negligible even when the multi-axis platform is

in motion. The results in Fig.5 show that $w$ averages 0 m s$^{-1}$ (uncertainty analysis in the next section), which indicates that the platform motion has successfully been removed. However, the standard deviation of $w$ (1-$\sigma$) increases notably with the rate of change of the pitch angle of the platform (Fig.6). Under the flight conditions reported in this work (Tables 2 and 3), the pitch angle rarely exceed $\pm 10$ deg s$^{-1}$ (Section 3.1). As the focus of the scientific research related to these wind measurements centers around updraft observations for aerosol-cloud interactions studies, Section 3.3 describes the uncertainty analysis related to the

calculation of vertical wind $w$.

## 3.3 Uncertainty analysis

The uncertainty analysis on the vertical wind vector $w$, Eq.(5), shows that the standard deviation, $\sigma_w$, is less than 0.1 m s$^{-1}$ based on wind tunnel experiments (Fig.5), with a lower limit of 0.05 m s$^{-1}$ for a fixed probe (no motion). As shown in Fig.6, the magnitude of $\sigma_w$ depends on the rate of angular change of the pitch angle, which suggests that $\sigma_w$ is related to a

convolution induced by the signal processing in the IMU that measure accelerations and angles. Figure 6 shows the $\sigma_w$ increases with higher frequency motion. In this section, we explore the parameters related to the uncertainty analysis for the vertical wind vector $w$. Equations (5) and (6) derive $\sigma_w$ using the individual parameters measured by the 5-hole probe and the IMU. From Eq.(1c), the propagation of errors associated with $w$ is:





$$\sigma_w = \sqrt{\left(\frac{\partial w}{\partial \alpha}\sigma_\alpha\right)^2 + \left(\frac{\partial w}{\partial V_a}\sigma_{V_a}\right)^2 + \left(\frac{\partial w}{\partial \theta}\sigma_\theta\right)^2} \tag{5}$$

where the relationship between measurement of differential pressure on the 5-hole probe and the corresponding $\alpha$ angle is given by:

$$\sigma_\alpha = a_\alpha \times \sigma_{C_\alpha} \tag{6a}$$

$$\sigma_{C_\alpha} = \sqrt{\left(\frac{\sigma_{\Delta(P_4-P_5)}}{\Delta(P_1-P_6)}\right)^2 + \left(\frac{\Delta(P_4-P_5)}{\Delta(P_1-P_6)^2} \times \sigma_{\Delta(P_1-P_6)}\right)^2} \tag{6b}$$

The uncertainties associated with each parameter from Eq.5 and Eq.6 are summarized in Table 1 and show that the lower limit for measuring updraft with this system is 0.11 m s$^{-1}$ corresponding to the observed measurements in Fig.6.

## 4 Comparison of 3D winds from RPAS and sonic anemometer

The comparisons between RPAS and sonic anemometer (Campbell CSAT3 3-D Sonic Anemometer) were conducted under calm wind conditions at Centre de Recherches Atmopheriques (CRA), which is an instrumented site of the Pyrenean Platform of Observation of the Atmosphere (P2OA), near Lannemezan, France. The purpose of the comparison is 1) to validate the measurement of 3D winds with RPAS by comparing Power Spectral Density (PSD) and vertical wind $w$ distributions, and 2) to calculate turbulent kinetic energy (TKE) used to study boundary layer dynamics and to compare with previous studies. Table 2 summarizes flight and wind conditions encountered during these validations. The sonic anemometers are installed on a meteorological mast at heights of 30 m.agl and 60 m.agl as part of permanent installations at the CRA. During the experiment, the RPAS flew straight N-S and E-W legs in the vicinity of the mast, at 60 m.agl in altitude. The leg length was 1600 m and the duration of flights was approximately 1.5 hours. A total of five flights were conducted at different times of the day and in different seasons: a series of three flights were conducted on 15 October 2015 at different times of the day, one flight was conducted in the morning on 20 May 2016, and the last flight was conducted in the afternoon on 7 July 2016 (Table 2). While all flights were conducted in low wind conditions, the turbulent conditions differed from one flight to another.

### 4.1 Validation of 3D wind measurements

Once the three components of the wind velocity from the RPAS are obtained with the Lenschow equations, Eq.(1), Power Spectrum Density (PSD) functions of RPAS are compared to PSD from sonic anemometer. In a well mixed boundary layer, PSD is expected to follow the -5/3 slope from Kolmogorov law. To facilitate comparison, PSD functions from legs and





anemometer were averaged (Fig.7). The PSDs transform the wind components into a frequency domain, and reveal the contribution of the RPAS in the wind velocity components. Note that frequencies lower than $10^{-2}$ Hz are sparse, therefore, the averaged values appear less smooth than higher frequencies. For flights 1, 2, 3 and 4, the RPAS motions are still visible at 0.1 Hz in the horizontal $u$-component due to inaccuracy of the heading measurement in the IMU. Flight 5 was conducted after

improved calibration of the heading measurement by the IMU, and a notable improvement in the PSDs (Fig.7; green line particularly on $u$-component) clearly demonstrates the importance of precise heading measurements. The PSDs of the wind velocities from both RPAS and anemometer follow the -5/3 slope as expected from the Kolmogorov law. We note, however, that the PSD from the RPAS is higher than the wind from anemometer, owing to the motion of the RPAS platform. The same trend has also been observed with a Manta UAV by Reineman et al. (2013) and a SUMO platform at Lannemezan in 2011

(Reuder et al., 2008).

To compare vertical wind from both platforms, angle of attack $\alpha$ and pitch angle $\theta$ from RPAS are centered such that average vertical velocity $w$ is 0 m s$^{-1}$ over the time of the flight. This step is needed because alignment of the 5-hole probe with reference to Earth's frame may change with meteorology conditions (such as wind speed) and flight parameters (such as

airspeed and nominal pitch angles). The re-centering step of the 5-hole probe on the RPAS is further supported by results from the sonic anemometer on the mast, which also show vertical velocity approaching 0 m s$^{-1}$ over the duration of the flight. Distributions of vertical wind velocity from RPAS and sonic anemometer are compared in Fig.8, and the intersection method is used to quantify the agreement between the distributions. For an exact match, the intersection method result is 1, for a complete mismatch, the result is 0. The calculation of the intersection number is obtained from $\sum_{i=1}^{N} min\left(I_i, M_i\right)$ with $I$ and

$M$ as the normalized distributions to be compared, and $N$ as the total number of bins. Table 2 summarizes the intersection numbers between the RPAS and anemometer vertical wind distributions for the five flights at P2OA, Lannemezan. The intersection numbers between RPAS and anemometer vertical wind distribution are higher than 70 % for 4 flights, indicating relatively good agreement between the RPAS and the sonic anemometer on the mast. The lowest intersection number (0.57) corresponds to flight 4, which is explained by the extra-motion induced on the RPAS due to an airspeed that was close to the

stall speed (ca. 12 m s$^{-1}$).

### 4.2   Turbulent kinetic energy

In the atmospheric boundary layer, the turbulent kinetic energy (TKE) quantifies the intensity of turbulence, which controls mixing of the atmosphere (Wyngaard and Coté (1971); Lenschow (1974)). TKE is defined as:

$$TKE = \frac{1}{2}(\sigma_u{}^2 + \sigma_v{}^2 + \sigma_w{}^2) \tag{7}$$

with $\sigma_u{}^2$ as the variance of E-W wind, $\sigma_v{}^2$ as the variance of N-S wind, and $\sigma_w{}^2$ as the variance of vertical wind. Previous results of TKE at P2OA, Lannemezan have been published for the BLLAST campaign, and TKE from the SUMO and M$^2$AV



have been compared to the sonic anemometer at 60 m.agl (Båserud et al. (2016); Lampert et al. (2016)). The reported values of TKE SUMO and M$^2$AV are within 50 % of TKE from the sonic anemometer. Canut et al. (2016) also showed a good agreement of TKE between a tethered balloon and the M$^2$AV, as well as, a manned aircraft. Correlation coefficients ($R^2$) between the tethered balloon and the sonic anemometer for the three variance components were between 0.8 and 0.9, while

the values of $R^2$ obtained with the RPAS in the present work are between 0.6 and 0.8 (note: fewer data points and less dynamic range for the RPAS measurements presented in this study). TKE calculated from the RPAS (TKE$_{RPAS}$) and the sonic anemometer on the mast (TKE$_{mast}$), for the five flights, are presented in Fig.9. The slope of the linear regression is 1.38 ($R^2 = 0.7$), which is similar to results reported by other studies at P2OA (Lampert et al. (2016); Båserud et al. (2016); Canut et al. (2016)).

To compare TKE$_{RPAS}$ and TKE$_{mast}$, a length scale is selected based on the length of the leg flown by the RPAS. The reference length scale for this study is 1600 m, which is the distance for each flight leg. The RPAS generally flies at an airspeed of 16 m s$^{-1}$, which translates to nearly 100 seconds per leg. For each flight, the time necessary for the anemometer to record wind traveling for each reference leg length is calculated using the horizontal wind speed at 30 or 60 m.agl (based on

data availability; Table 2). The time interval used to calculate TKE$_{mast}$ is centered on the RPAS leg. TKE is calculated leg-by-leg before being averaged. For example, the horizontal wind velocity during flight 5 is 1.7 m s$^{-1}$; thus, TKE$_{mast}$ is calculated based on the time required for the air to travel 1600 m, which is 940 s. After obtaining a value of TKE for each leg, the values are averaged to show a single TKE over the entire flight for both the mast and RPAS. In Fig.9, the error bars associated with TKE (x-axis) represent minimum and maximum values of TKE$_{mast}$ associated with the different length scales.

To study the influence of length scale on the calculation of TKE$_{mast}$, as well as the isotropy in the N-S and E-W directions, variances of the horizontal wind components from the mast are obtained from 800 m to 6400 m based on the horizontal wind speed travel time (as stated above). The variances on $u$ (E-W wind component), and $v$ (N-S wind component), are shown in Fig.10. At 30 m.agl, the variances ($\sigma_{u,mast}{}^2$ and $\sigma_{v,mast}{}^2$) increase with the length scale, creating a range in variances

between 0.1 to 0.4 m$^2$ s$^{-2}$ depending on the flight. However, at 60 m.agl, the range in variances remains below 0.1 m$^2$ s$^{-2}$ for $\sigma_{u,mast}{}^2$ and 0.15 m$^2$ s$^{-2}$ for $\sigma_{v,mast}{}^2$ for all length scales. The isotropy between the N-S and E-W directions is confirmed when variances in $u$ and $v$ are similar. At 30 m.agl, Fig.10 shows that $\sigma_{v,mast}{}^2$ is up to a factor of two higher than $\sigma_{u,mast}{}^2$; therefore isotropic conditions are not satisfied at 30 m.agl. While at 60 m.agl, $\sigma_{u,mast}{}^2$ and $\sigma_{v,mast}{}^2$ are within 15 %, implying isotropy for altitudes in the boundary layer above 60 m.agl. Differences observed in variances between the

horizontal wind components at 30 m.agl are related to surface topography (e.g., nearby trees, fields...).

While isotropic conditions are satisfied for measurements in the boundary layer above 60 m.agl, based on mast observations, these expected conditions are not observed between the transversal direction (normal to the axis of the RPAS fuselage) and longitudinal direction (parallel to the axis of the RPAS fuselage) based on the RPAS measurements. Therefore,

to calculate TKE$_{RPAS}$, we assume isotropy, where $\sigma_{u,RPAS}{}^2$ equals $\sigma_{v,RPAS}{}^2$. Figure 9 illustrates the impact of the isotropy



assumption on the calculation of $TKE_{RPAS}$ values. The increase in variances in the transversal wind direction is related primarily to uncertainties in the IMU measurement of the RPAS heading relative to the Earth coordinate system. Note when the RPAS is flying along N-S legs, the transversal wind in RPAS coordinate system is the E-W wind component, likewise, the transersal wind in the N-S wind component corresponds to flights along E-W legs. To conclude, we use the isotropy

assumption to calculate $TKE_{RPAS}$ based on the variance in the longitudinal direction (i.e., parallel to the axis of the RPAS fuselage).

In Fig.9, the TKE calculated for flight 4 is significantly different from the other flights as the RPAS was close to stall speed (as mentioned in Section 4.1, flight 4 also yielded the lowest intersection number). While beyond the scope of this paper,

these results suggest that improving the measurement of horizontal winds and reducing biases in the horizontal components of the variances may be achieved by 1) improvement of the IMU heading measurement (also noted in Elston et al. (2015)), and 2) verified with a flight plan in a cross pattern (i.e., orthogonal legs).

## 5   Comparison of vertical wind velocities from RPAS and cloud radar

A BACCHUS field campaign took place at the Mace Head Atmospheric Research Station on the west coast of Ireland in August 2015. The purpose was to study aerosol-cloud interactions linking ground-based and satellite observations using RPAS (Sanchez et al., 2017). Among the four instrumented RPAS which flew at Mace Head, the wind-RPAS was equipped with a 5-hole probe and an IMU to obtain 3D wind vectors, as well as upward and downward facing pyranometers to measure downwelling and upwelling broadband solar irradiance (400 to 1100 nm wavelengths). During the campaign, we concentrated

on measurements of vertical wind velocity near cloud base to study aerosol-cloud interactions. After identifying the cloud base from the ceilometer or a vertical profile of an earlier flight, the wind-RPAS was sent to an altitude close to cloud base, flying 6 km-long straight-and-level legs. Horizontal wind speeds varied from 6 to 12 m s$^{-1}$ from the West during the case studies presented here. The presence of clouds during the flight was determined using the ratio of the upwelling to downwelling solar fluxes; when the RPAS was underneath or within a cloud, the ratio approaches unity. During this field

campaign, the wind-RPAS flew in 10 of the 45 scientific flights for a total of 15 hours. Here, we focus on three flights with the wind-RPAS (Table 3), in which the vertical wind is compared to vertical wind from the cloud radar at Mace Head (Cloud Radar MIRA-35, METEK). Of the 10 flights with the wind-RPAS, three provided reliable measurements of updraft inside the clouds, while the other flights were not selected for a number of reasons: water in the 5-hole probe (2 flights), insufficient number of cloud radar data for comparison (2 flights), no cloud (1 flight), no pyranometer data to identify clouds (1 flight),

and aborted mission due to strong winds (1 flight). The Doppler cloud radar (35.5-GHz, Ka band) is adapted to the observation of the cloud structure over the whole vertical range (Görsdorf et al., 2015). The cloud radar is equipped with a vertically pointed antenna with a polarization filter, a magnetron transmitter and two receivers for polarized signals. Measurements are available up to 15-km height for a temporal resolution of 10 seconds. In this study, a single range bin of the



cloud radar is selected for the comparison; either cloud top or RPAS flight altitude. The vertical resolution of the cloud radar is 29 m. While the flight altitude for the wind-RPAS was estimated to be near the cloud base, uncertainties in retrieving cloud base height or an evolution in cloud base height related to diurnal cycles of the boundary layer inevitably lead to the possibility of the RPAS flying in the clouds rather than just below cloud base. Consequently, data from two wind-RPAS flights

during the field campaign are not presented due to the accumulation of water in the probe. We are currently improving the instrumentation to address the issue. Note that direct comparison of instantaneous data between the RPAS and cloud radar was not possible, as the RPAS did not fly directly over the cloud radar, and did not observe the same air mass. Moreover, the cloud radar reports vertical velocities every 10 s (and only when a cloud is present); therefore relatively long averaging preriods are needed to compare with the RPAS observations. Hence, we present selected time series of the cloud radar

measurements that represent the state of the atmosphere during the flight; for cases with sufficient cloud cover, we present different averaging periods of the cloud radar (a short period that coincides with the RPAS flight and a long periods for better counting statistics). Normalized distributions are plotted on the same interval divided into 30 bins of vertical wind velocities. In the present study, comparisons of the vertical wind velocity of the cloud radar serve to validate the RPAS results, as well as to provide insight on different atmospheric states related to the measurement techniques.

### 5.1   Stratocumulus deck with light precipitation (Flight 26: 2015/08/11)

On 11 August, the sky was covered by a stratocumulus deck, and the wind-RPAS flew at 1160 m.asl (at least 100 m above the cloud base). Time series of vertical wind from the cloud radar are presented in Fig.11, along colored horizontal lines that indicate observation periods of the cloud radar and RPAS, and Fig.12 presents a comparison of the vertical wind distribution

obtained by cloud radar measurements and the RPAS flight. The standard deviation of RPAS vertical wind distribution is $\sigma_{RPAS} = 0.19$ m s$^{-1}$ (or 0.11 m s$^{-1}$ if only positive vertical velocities are considered). This result is within the range of vertical wind standard deviations obtained in Lu et al. (2007) for stratocumulus clouds observed off the coast of Monterey, California, eastern Pacific. In this case study, the presence of falling cloud droplets to an altitude as low as 300 m.agl (Fig.11) negatively biases the vertical wind distribution of the cloud radar (Fig.12). Previous measurements have also shown that precipitation

negatively biases cloud radar observations of vertical wind velocities, as the radar indirectly measures vertical wind by using the motion of scatterers (i.e., hydrometeors; Lothon et al. (2005), Bühl et al. (2015)). These negative biases related to the falling drops are largely removed by obtaining vertical velocity at the top of the cloud (Bühl et al., 2015). Similar results are obtained for our case study, as the cloud radar is strongly influenced by falling droplets, yet only slightly negatively biased at the cloud top. The intersection method, described in Section 4.1, is used to compared the three radar vertical wind

distributions with the distribution of RPAS measurements (Table 3). The intersection number of 0.53 between radar flight altitude and RPAS vertical wind distributions confirms the low match as a result of the negative bias from the precipitating droplets. However, a much better agreement is found between the cloud radar vertical velocity at the top of the cloud (1360 m.asl) and the RPAS measurements (intersection number = 0.74).



## 5.2 Cloud fields within changing meteorology (Flight 38: 2015/08/21)

A comparison of results from the RPAS and cloud radar emphasizes the differences in vertical winds depending on the regions within the cloud field. During Flight 38, the RPAS flew within a cloud above the ocean and in clear sky above land for three legs, after which the local meteorology changed into a formation of developing clouds above land (where a cloudless sky had previously been observed; Fig.13). The vertical wind velocity for the Flight 38 is presented using a combination of information shown in a series of figures: downwelling and upwelling pyranometer observations, and three periods corresponding to distinct meteorological conditions (Fig.13); the time series cloud radar data (Fig.14); and the vertical wind distributions from the RPAS flight and the cloud radar (Fig.15). These meteorological periods are defined in Fig.13 as "cloud" (both pyranometers approach similar values), "no cloud" (downwelling pyranometer is significantly higher than upwelling pyranometer), and a third period associated to a developing field of broken clouds (spatially variable downwelling pyranometer). Combining information from Figs.13, 14, and 15, we deduce a cloudless sky (cyan) was observed by the RPAS above land for the first three legs (Fig.13). The corresponding cloud radar time series also showed a cloudless sky above land for the beginning of the flight (Fig.14). In the meantime, the RPAS flew within a cloud above the ocean (green), which was not observed by the cloud radar. Figure 15 shows that the standard deviation of vertical velocity within the cloud is larger than for clear sky conditions ($\sigma_{cloud}$ = 0.29 m s$^{-1}$, $\sigma_{nocloud}$ = 0.17 m s$^{-1}$) , which highlights the presence of stronger vertical winds in the presence of clouds. During the last two legs of Flight 38, the wind-RPAS flew through a developing field of broken clouds above land (magenta), which also appeared in the cloud radar time series and in the satellite image (Fig.4 in Sanchez et al. (2017)). The standard deviation of the "broken cloud" RPAS period of vertical wind is larger than the other periods ($\sigma_{broken\,cloud}$ = 0.48 m s$^{-1}$). Similar vertical wind distributions are found for cloud radar and the RPAS during the "broken clouds" period (Fig.15). While not shown here, the vertical wind distributions observed by the cloud radar are similar at radar cloud base (380 m.asl) and at the flight altitude (660 m.asl), as well as at different observing periods (1.5 and 4 hours). In Table 3, intersection numbers illustrate the relatively close matches (ca. 80 %) in comparing the "broken cloud" RPAS period and the cloud radar for 4 hours (radar flight altitude) and for 1.5 hours (radar flight time). For comparison, the values of intersection numbers between "broken cloud" and "cloud" periods is 0.73, while between "broken cloud" and "no cloud" periods is 0.56 (based on RPAS measurements). The similar results for the observations of a field of broken clouds independently reinforces RPAS and cloud radar observational methods, and the changes meteorological conditions highlight the ability to identify distinct states of the atmosphere with the RPAS. Relating these differences in updraft velocity to the meteorological conditions of the boundary layer will be explored in future studies.

## 5.3 Fair weather cumulus clouds (Flight 30: 2015/08/15)

During Flight 30, the cloud field was scattered with small clouds as shown in Fig.17 by the cloud radar time series. The wind-RPAS flew through one of these clouds as shown by the pyranometer measurements in Fig.16. However, the number of data points from the cloud radar during the flight time (black segment) was insufficient to establish a vertical wind





distribution, therefore only the cloud radar data for 4 hours (red segment) are presented in Fig.18. To compare cloud radar and RPAS data, vertical winds from the RPAS are divided into "cloud" and "no-cloud" periods based on pyranometer observations. The respective standard deviations for the periods are $\sigma_{cloud} = 0.37$ m s$^{-1}$ and $\sigma_{nocloud} = 0.36$ m s$^{-1}$, which are not statistically different. However, the variability between legs is significantly greater in the "no cloud" period (as

represented by the envelope in blue dashed lines in Fig.18) compared to the "cloud" period (envelope in green dashed lines). In Fig.18, the RPAS and cloud radar measurements show similar results during the "cloud" period, with an intersection number equal to 0.78. Kunz and de Leeuw (2000) have observed an upward component in the air flow from the ocean at the Mace Head Research Station as a result of the terrain. There were less significant tilt angle effect on the wind direction at 22 m.asl. However, systematic differences between the RPAS and cloud radar have not been observed for the other case studies,

so we cannot quantify the role of surface heating or orography on the cloud radar vertical distributions compared to those of the RPAS. Nonetheless, Ansmann et al. (2010) have observed asymmetry in the vertical wind distributions related to the spatial distribution of the cloud field.

## 6    Conclusions

The validation of 3D wind measurements measured by a 5-hole probe on a lightweight remotely piloted aircraft system (RPAS) has been detailed in this study. The 5-hole probe has been calibrated in wind tunnel on a dynamic platform to obtain the angle of attack, angle of sideslip and airspeed of the RPAS. Motions induced by the two-axis platform in the wind-tunnel were effectively removed, thereby validating sensor performance. With an inertial measurement unit (IMU) providing ground speeds, Euler angles and GPS coordinates, 3D wind vectors have been calculated with the simplified wind equations from

Lenschow and Spyers-Duran (1989). The uncertainty associated with the vertical wind measurement has been determined to be 0.11 m s$^{-1}$. The 3D wind vectors from the RPAS showed good agreement with results from a sonic anemometer on a 60 m.agl meteorological tower at P2OA, Lannemezan, France. Vertical velocity distributions were compared from both platforms, and showed intersection values higher than 70 % in calm wind conditions. Comparisons have also been made on the power spectral density (PSD) functions between the sonic anemometer and RPAS measurements, which in both cases follow the Kolmogorov

law for established turbulent regime. In order to calculate the turbulent kinetic energy (TKE) parameter, the isotropy assumption ($\sigma_u{}^2 = \sigma_v{}^2$) has been applied on the horizontal wind from RPAS in order to correct biases in measurements resulting from heading inaccuracy. In the future, heading measurements will be improved with an IMU that includes differential GPS antennas installed in the RPAS. TKE value from sonic anemometer and RPAS agreed within 50 % (within other values reported in the literature), and showed the expected evolution of TKE turbulence. Three case studies from a BACCHUS field campaign (at

Mace Head Atmospheric Research Station, Ireland) validated RPAS vertical wind velocities near cloud base compared to cloud radar observations. Vertical wind velocity distributions were classified according to the flight periods (clear sky or cloud), emphasizing the impact of meteorology and the state of the atmosphere on the clouds. For the first case study, a stratocumulus





deck covered the sky and light precipitation was observed. Cloud radar vertical wind velocity distribution was negatively biased and cloud base was not distinctly visible due to falling droplets. The wind-RPAS provided a centered vertical wind distribution near cloud base, which was similar to cloud radar observations at cloud-top (in the non-precipitating region of the cloud). The second case study displayed different meteorological conditions during the flight, which were well distinguished by the wind-

RPAS, including differences between a developing field of broken clouds, a small convective cloud and clear sky. In the third case study, similar vertical wind distributions were observed near cloud base by the RPAS and the cloud radar in fair weather cumulus cloud systems above land and ocean. The distinct meteorological conditions which were encountered for each of the case studies validated RPAS results compared to cloud radar and highlighted the ability of the RPAS platform to differentiate cloud systems based on vertical wind measurements for different conditions. Vertical wind velocities near cloud base measured

in this study have been implemented in air parcel models to conduct studies of aerosol-cloud interactions closure study with ground-based measurements and satellite observations (Sanchez et al., 2017).

*Acknowledgements.* The research leading to these results received funding from the European Union's Seventh Framework Programme (FP7/2007-2013) project BACCHUS under grant agreement n°603445. NUI Galway was also supported by the HEA under PRTLI4, the EPA, and SFI through the MaREI Centre. Part of the remotely piloted aircraft system operations presented here have been conducted at

Centre de Recherches Atmosphériques of Lannemezan (an instrumented site of the Pyrenean Platform of Observation of the Atmosphere, P2OA), supported by the University Paul Sabatier, Toulouse (France) and CNRS INSU (Institut National des Sciences de l'Univers). The RPAS used for the experiments presented in this work have been developed by the Ecole Nationale de l'Aviation Civile (ENAC). We also thank Marie Lothon from P2OA Lannemezan, Bruno Piguet and Guylaine Canut from Centre National de Recherches Météorologiques (CNRM) for their advices on wind measurements, and Joachim Reuder and Line Båserud from the Geophysical Institute, Univeristy of

Bergen, Norway, for their advices on TKE calculation.



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





**Table 1.** Uncertainty (1-$\sigma$) associated with parameters from 5-hole probe (5HP) and inertial measurement unit (IMU) for the calculation of error associated with the vertical wind velocity $w$.

| Variable | Symbol | precision/value |
|---|---|---|
| differential pressure between holes 1 and 6 (5HP) | $\sigma_{\Delta(P_1-P_6)}$ | 0.012 mbar |
| differential pressure between holes 4 and 5 (5HP) | $\sigma_{\Delta(P_4-P_5)}$ | 0.012 mbar |
| ratio of differential pressures (5HP) | $\sigma_{C_\alpha}$ | 0.012 |
| angle of attack (5HP) | $\sigma_\alpha$ | 0.19 deg |
| pitch angle (IMU) | $\sigma_\theta$ | 0.03 deg |
| airspeed (5HP) | $\sigma_{V_a}$ | 0.1 m s$^{-1}$ systematic 7 % |
| vertical ground speed (IMU) | $\sigma_{V_p}$ | 0.1 m s$^{-1}$ |
| coefficient calibration - slope | $a_\alpha$ | 15.06 |
| coefficient calibration - intersect | $b_\alpha$ | -0.78 |
| calculated vertical velocity | $\sigma_w$ | 0.11 m s$^{-1}$ |

**Table 2.** Description of flights conducted at P2OA, Lannemezan, France.

| ID | Date | Time (local) | Duration | Horizontal wind speed | Wind direction | Intersection number* | Remarks |
|---|---|---|---|---|---|---|---|
| Flight 1 | 15 Oct 2015 | 08:05 | 1h30 | 0.6 m s$^{-1}$ | NE | 0.71 | sonic anemometer at 30 m.agl only |
| Flight 2 | 15 Oct 2015 | 12:47 | 1h22 | 1.9 m s$^{-1}$ | N | 0.78 | sonic anemometer at 30 m.agl only |
| Flight 3 | 15 Oct 2015 | 15:35 | 1h18 | 2.7 m s$^{-1}$ | NW | 0.88 | sonic anemometer at 30 m.agl only |
| Flight 4 | 20 May 2016 | 09:15 | 1h35 | 3.2 m s$^{-1}$ | SW | 0.57 | airspeed close to stall speed |
| Flight 5 | 7 Jul 2016 | 15:18 | 1h06 | 1.7 m s$^{-1}$ | NE | 0.86 | magnetometer re-calibrated |

*intersection number described in Section 4.1



**Table 3.** Description of BACCHUS case study flights, Mace Head, Ireland. The intersection number compares vertical wind velocity distributions between RPAS and cloud radar.

| ID | Date | Time (local) | Duration | Horizontal wind speed | wind direction | Intersection number*** (comparison with RPAS) | | | Figure |
|---|---|---|---|---|---|---|---|---|---|
| | | | | | | radar flight alt. | radar flight time | radar cloud top | |
| Flight 26* | 11 Aug 2015 | 16:17 | 1h20 | 6 m s$^{-1}$** | WNW to SW | 0.53 | 0.67 | 0.74 | Fig.12 |
| Flight 30 | 15 Aug 2015 | 14:19 | 50 min | 10 m s$^{-1}$** | W to WSW | 0.78 | | | Fig.18 |
| Flight 38 | 21 Aug 2015 | 16:10 | 1h30 | 10 m s$^{-1}$** | SSW | 0.82 | 0.78 | | Fig.15 |

*no pyranometer data

**the uncertainty associated with RPAS horizontal wind is $\pm 2$ m s$^{-1}$

***intersection number described in Section 4.1





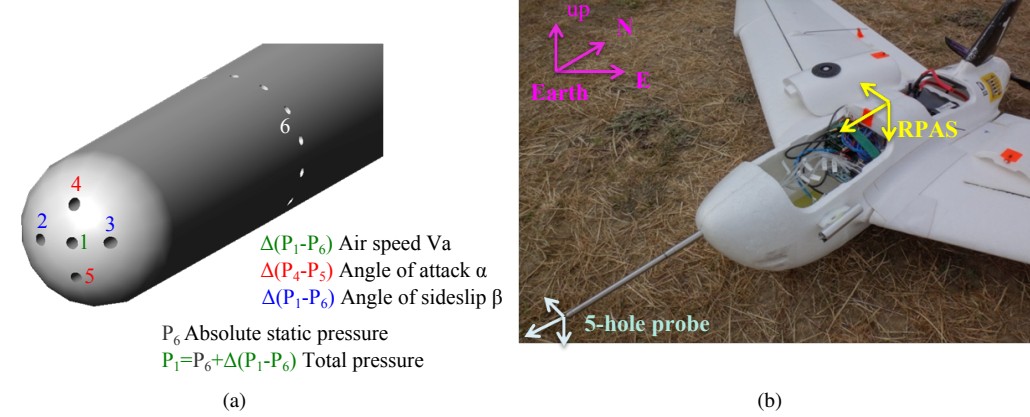

Figure 1. (a) 5-hole probe tip, schematic representation of pressure holes. (b) 5-hole probe mounted on a Skywalker X6 RPAS.

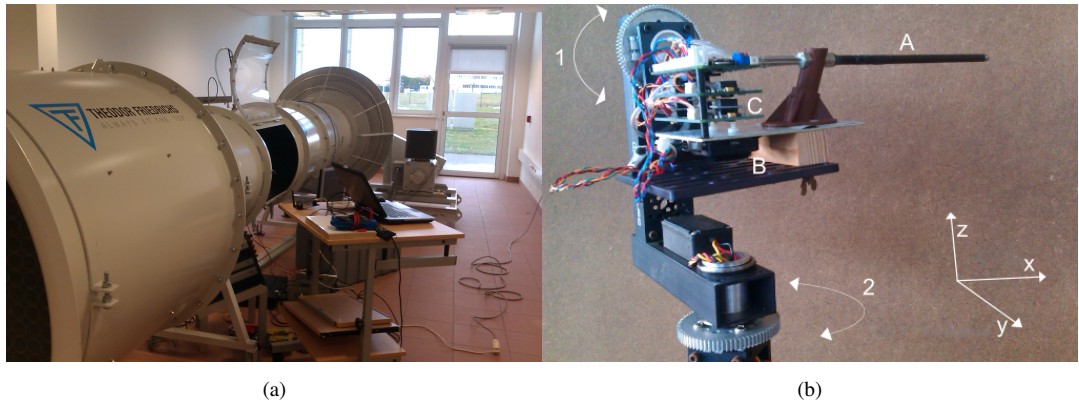

Figure 2. (a) Theodor Friedrichs wind tunnel, Météo-France, Toulouse. (b) Nacelle for wind tunnel set up, 1: rotation on pitch axis, 2: rotation on yaw axis, A: 5-hole probe, B: IMU, C: pressure sensors .





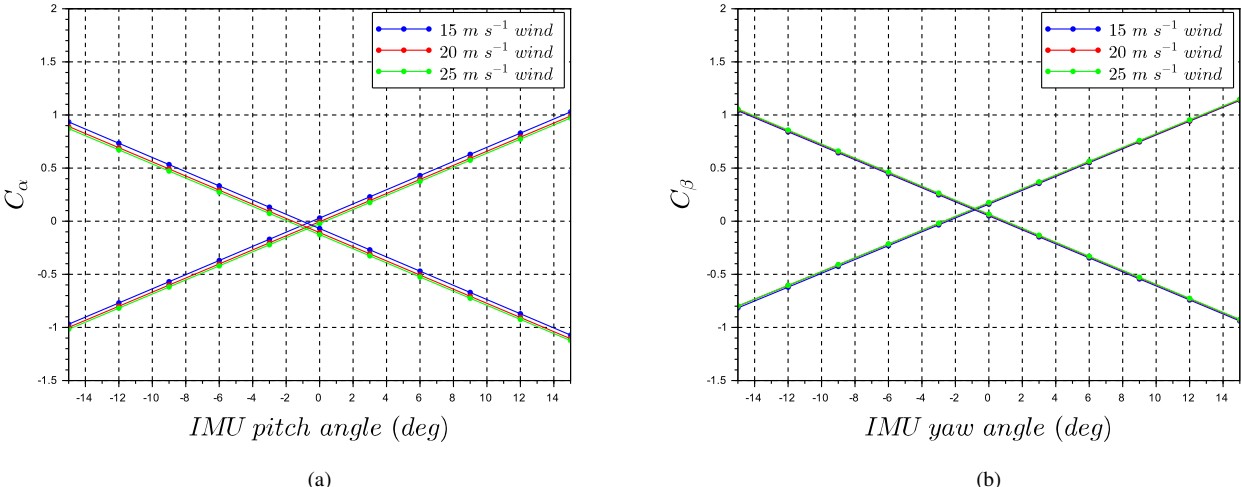

(a)  (b)

**Figure 3.** (a) Calibration coefficients $C_\alpha$ for given IMU pitch angles at different wind velocities, the positive slope corresponds to the probe in standard orientation, the negative slope corresponds to the probe in inverted orientation. (b) Calibration coefficients $C_\beta$ for given IMU yaw angles at different wind velocities, the positive slope corresponds to the probe in standard orientation, the negative slope corresponds to the probe in inverted orientation.

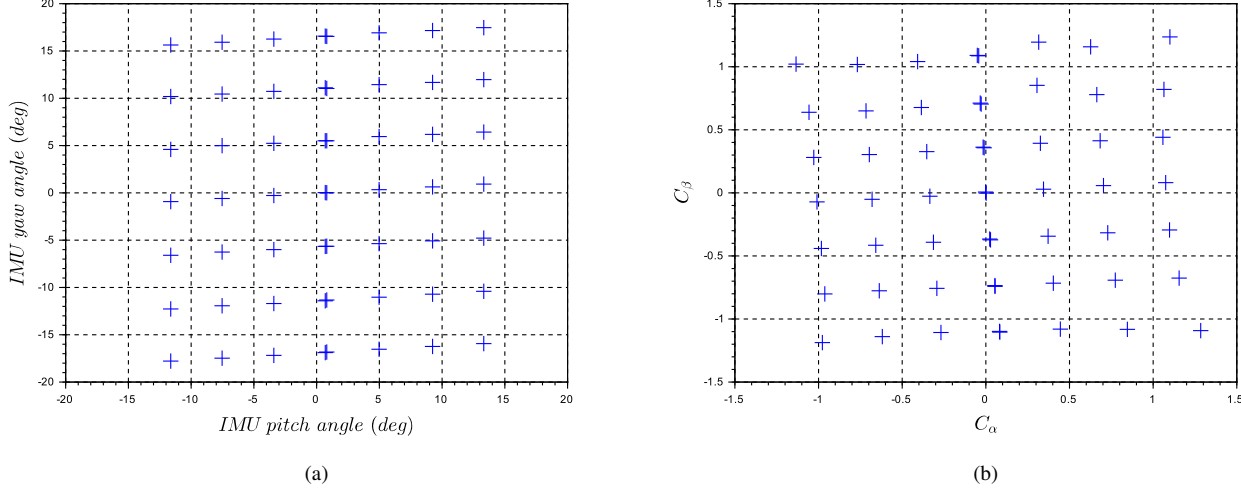

(a)  (b)

**Figure 4.** (a) Variation by step of pitch and yaw angles of the multi-axis platform in wind tunnel, wind speed 15m s$^{-1}$. (b) Corresponding $C_\alpha$ and $C_\beta$ of the 5-hole probe to steps of the platform.





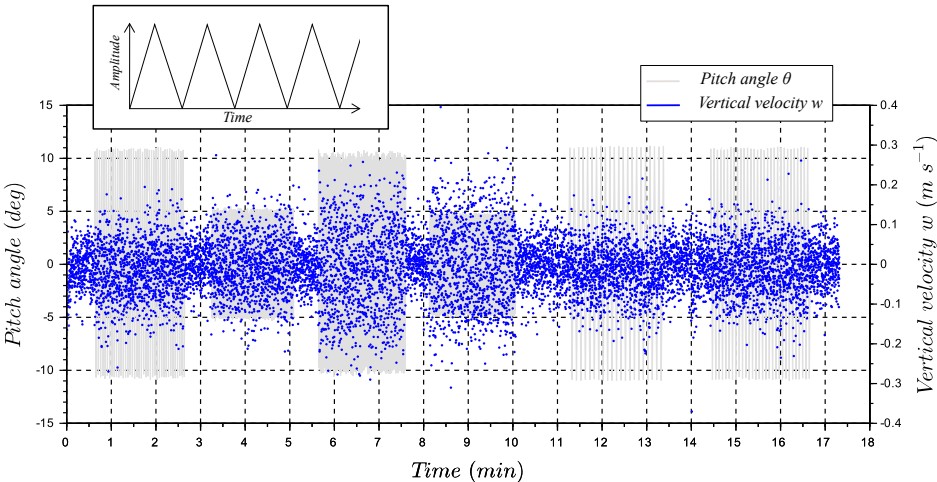

**Figure 5.** (Main figure) Calculated vertical wind vector $w$ for multi-axis platform motions in wind tunnel (blue) for different pitch angles and rates of motion (gray). (Inset) Expanded view of triangular motion applied to the pitch axis of the platform.

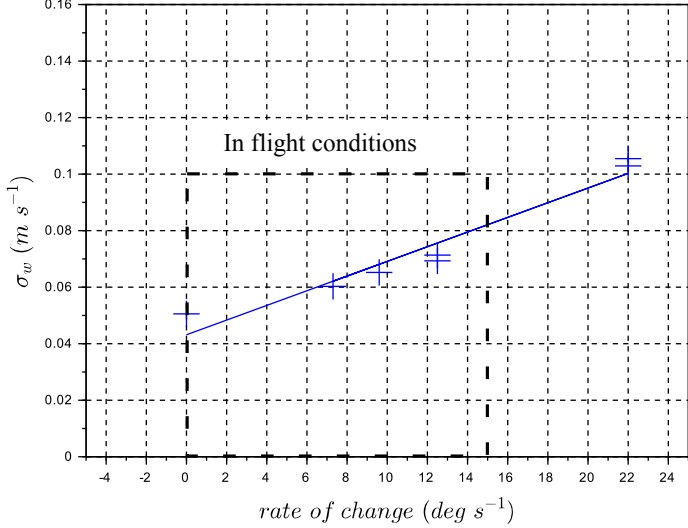

**Figure 6.** Standard deviation of vertical wind vector $w$ for each rate of change of the pitch angles on the multi-axis platform (Fig.5). For the flight conditions presented in this study, the rate of change does not exceed 15 deg s$^{-1}$.





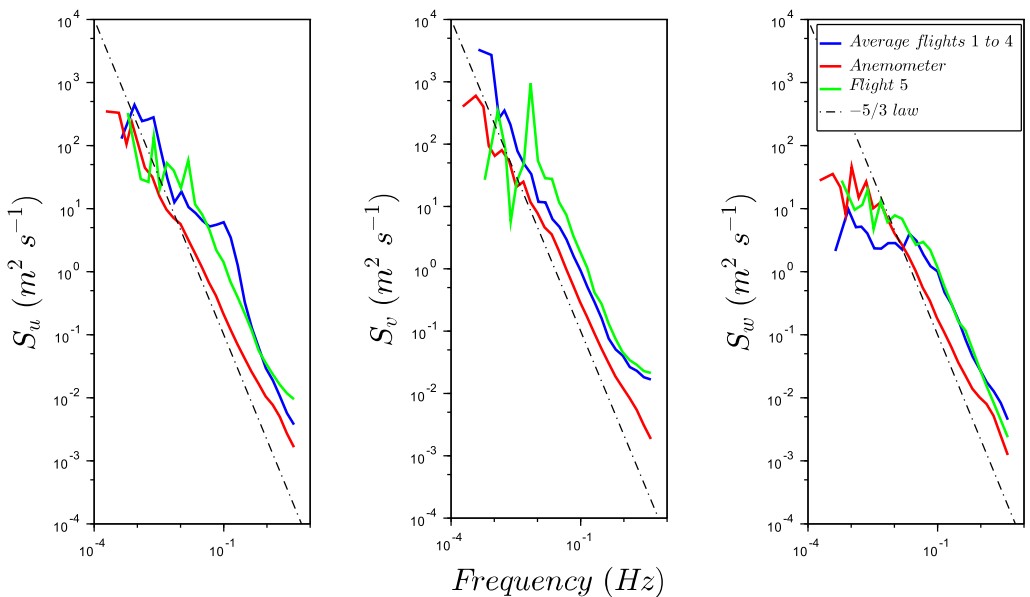

**Figure 7.** Comparison of averaged PSD from sonic anemometer and RPAS wind measurements for four flights, and flight 5 after re-calibration of the magnetometer. Each sub-figure corresponds to the spectral energy $S$ of wind components — $u$ (left), $v$ (middle), and $w$ (right) — function of frequency. The dashed line represents the $f^{-5/3}$ law.

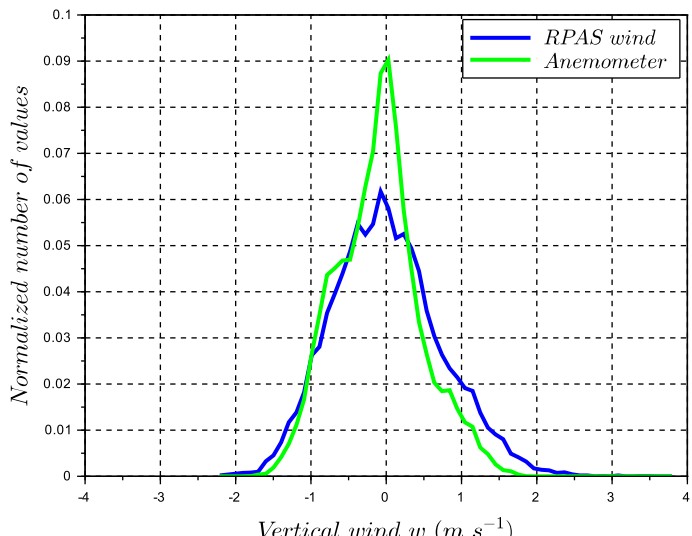

**Figure 8.** Distribution functions of vertical wind $w$ for RPAS and sonic anemometer measurements, flight 5.




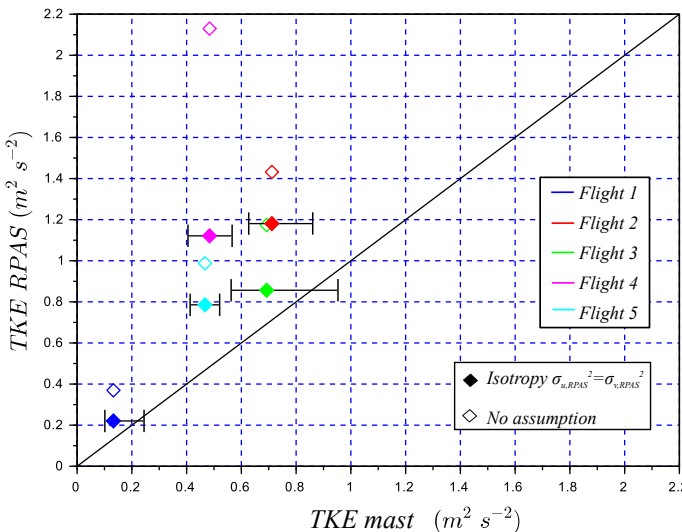

**Figure 9.** Comparison of TKE$_{RPAS}$ and TKE$_{mast}$. Open diamonds TKE$_{RPAS}$ are calculated with the three wind variances of RPAS, solid diamonds are obtained with the isotropy assumption $\sigma_{u,RPAS}^2 = \sigma_{v,RPAS}^2$. TKE$_{mast}$ is calculated with 1600 m length scale, the associated uncertainty bars cover length scales from 800 m to 6400 m.

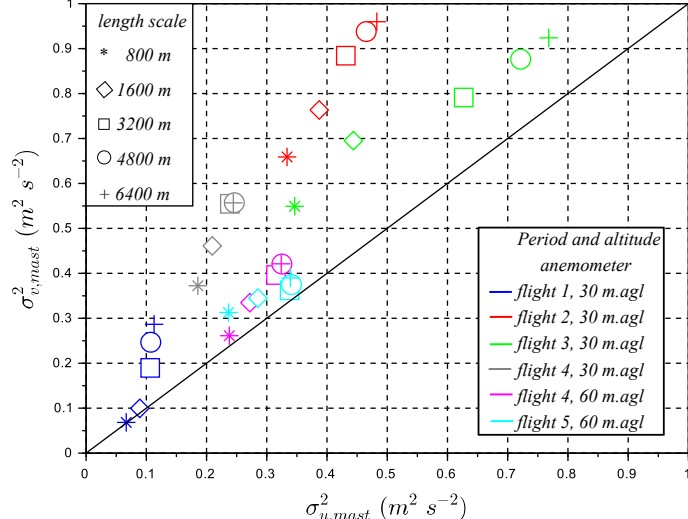

**Figure 10.** Comparison of variances $\sigma_{u,mast}^2$ and $\sigma_{v,mast}^2$ from the sonic anemometer for the associated flight periods.





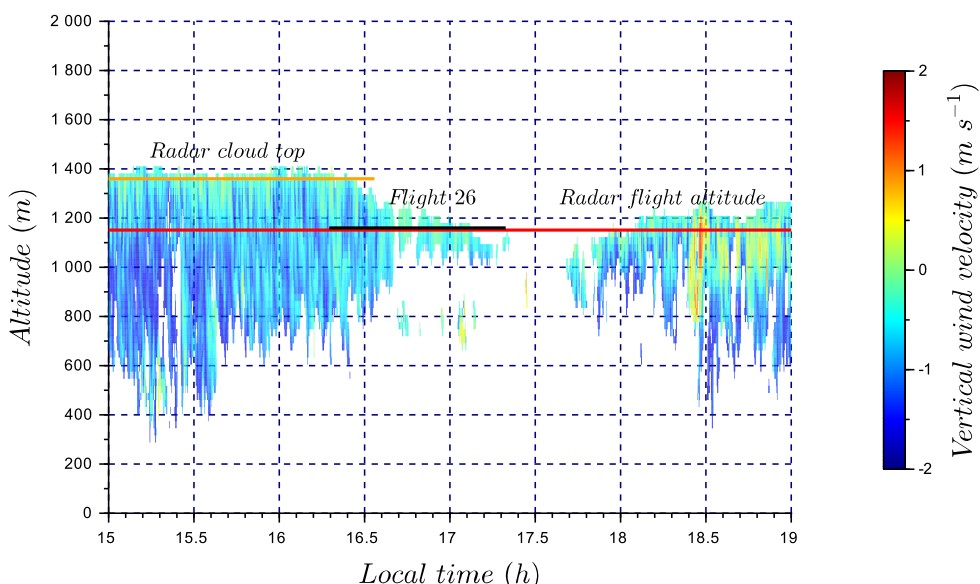

**Figure 11.** Time series of vertical wind velocity for a stratocumulus deck with light precipitation (Flight 26). The color bar represents the cloud radar vertical wind velocity. Flight 26 sampling time is identified by the black segment. The red horizontal line corresponds to radar data at the altitude of the RPAS (1160 m.asl), and the orange line corresponds to radar data at cloud top (1360 m.asl). Cloud radar vertical wind velocity at 1160 m.asl and 1360 m.asl are used in Fig.12 to compare with RPAS measurements.




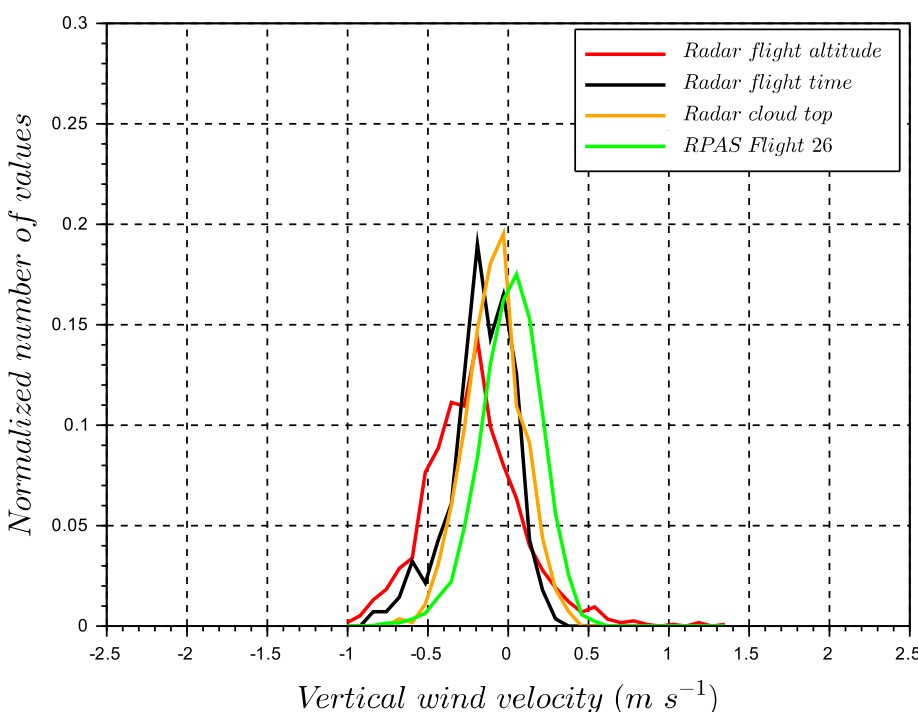

**Figure 12.** A comparison of vertical wind velocity distributions in a lightly precipitating stratocumulus deck between RPAS (1160 m.asl) and cloud radar at RPAS altitude (1160 m.asl) for 4 h and flight time periods, and cloud radar at cloud top (1360 m.asl).



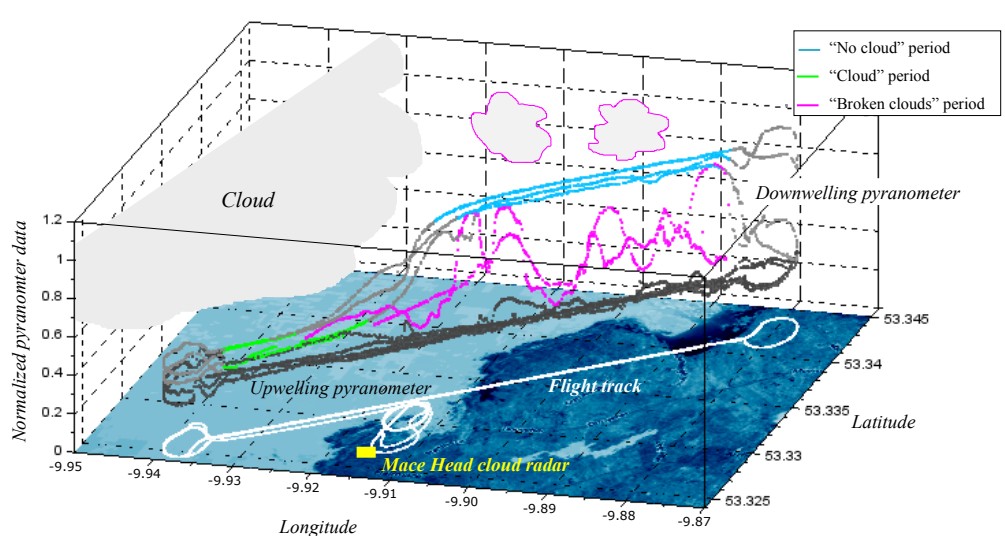

**Figure 13.** Coastal map and flight tracks for the case study of a convective cloud with changing meteorology (Flight 38). Downwelling and upwelling pyranometers data are color-coded based on the three flight periods ("cloud", "no cloud" and "broken clouds"). The developing field of broken clouds (magenta contour clouds) appeared during the last two legs. The cloud radar (yellow square) operated at the Mace Head research station.





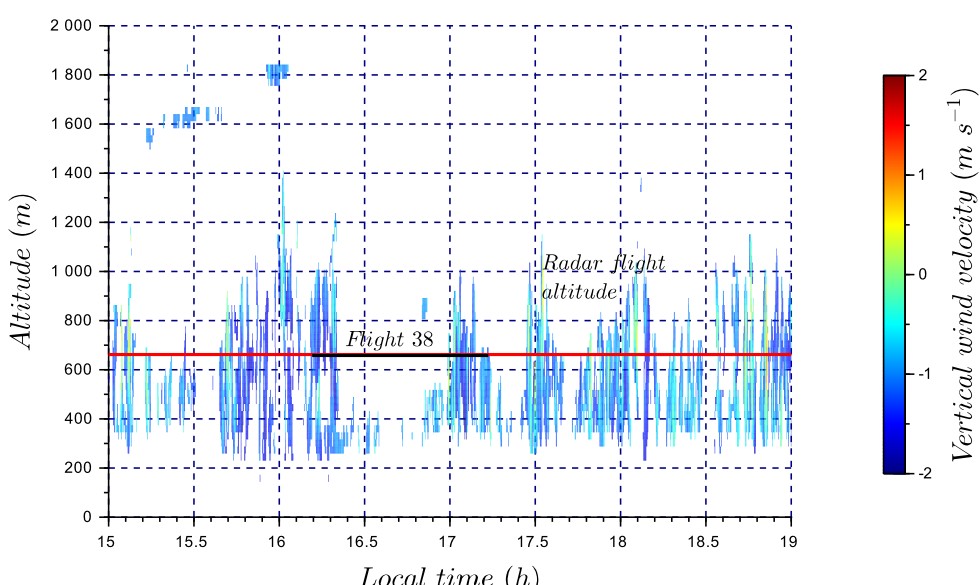

**Figure 14.** Time series of vertical wind velocity associated with Flight 38. The color bar represents the cloud radar vertical wind velocity. Flight 38 sampling time is identified by the black segment. The red horizontal line corresponds to radar data at flight altitude (660 m.asl), used to plot vertical wind velocity distributions in Fig.15.





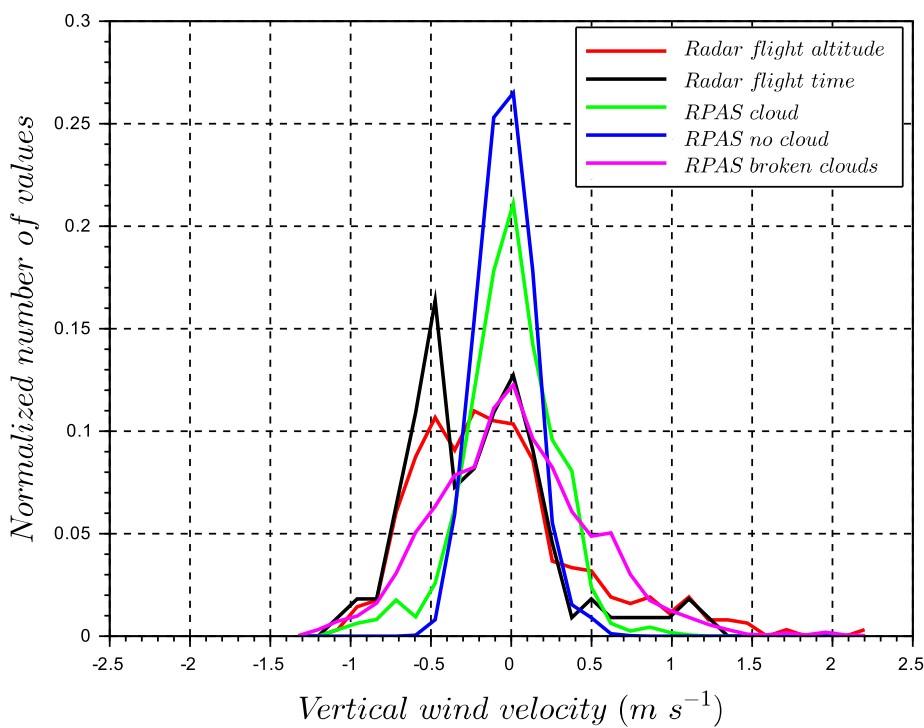

**Figure 15.** Comparison of vertical wind velocity distributions for RPAS and cloud radar for Flight 38 at RPAS flight altitude from Fig.14. RPAS measurements are divided into periods defined in Fig.13 ("cloud", "no cloud" and "broken clouds" periods). The cloud radar detected cloud only for the "broken clouds" period during the RPAS flight.





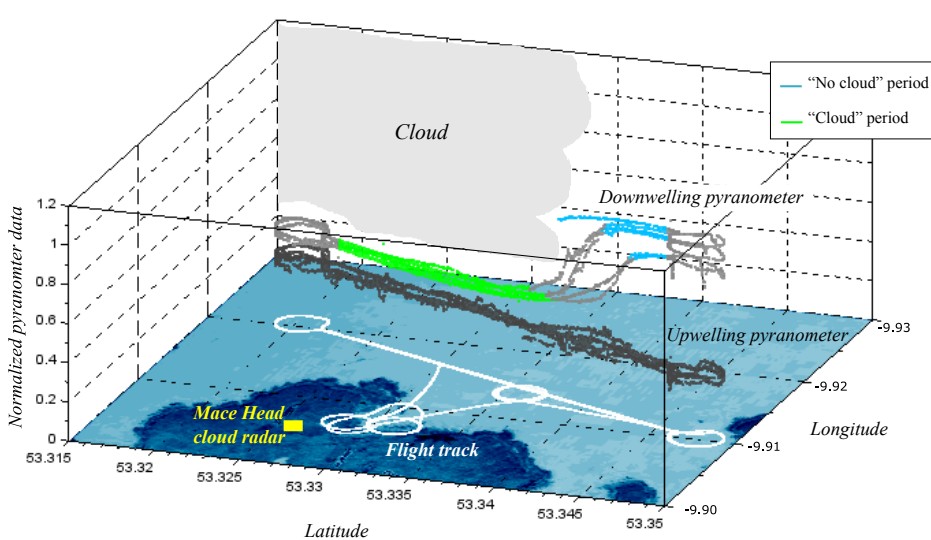

**Figure 16.** Coastal map and flight tracks for the non-convective cloud case study (Flight 30). Downwelling and upwelling pyranometers data are color-coded based on two flight periods, "cloud" and "no cloud" periods. The cloud radar (yellow square) operated at the Mace Head research station.





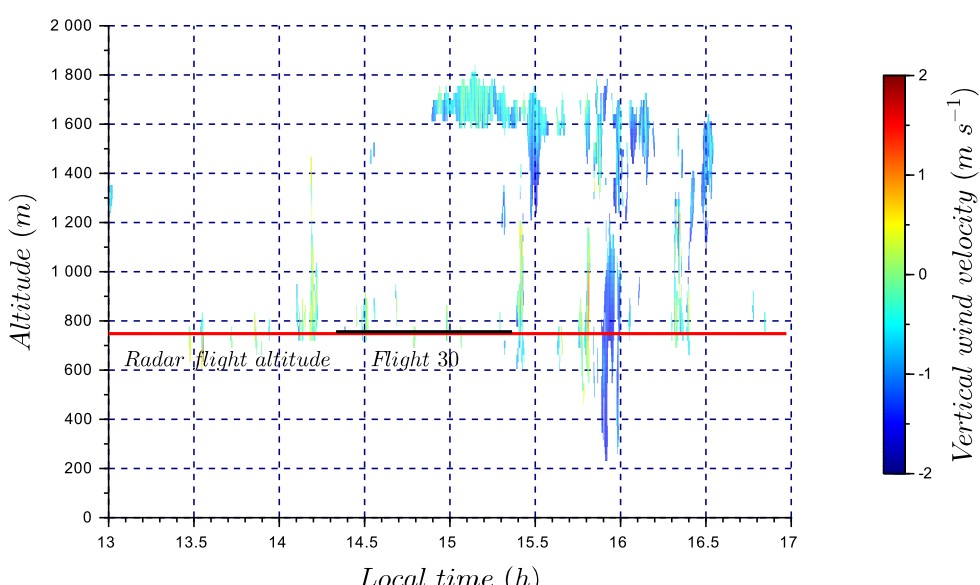

**Figure 17.** Time series of vertical wind velocity associated with Flight 30. The color bar represents the cloud radar vertical wind velocity. Flight 30 sampling time is identified by the black segment. The red horizontal line corresponds to radar data at flight altitude (750 m.asl) used in Fig.18 to plot vertical wind velocity distribution.





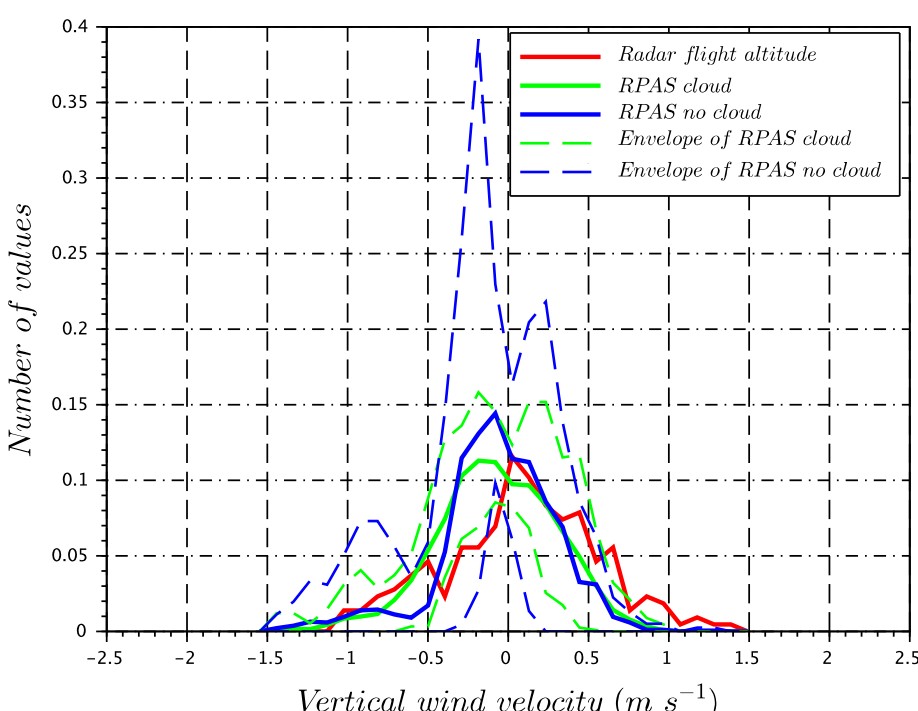

**Figure 18.** Comparison of normalized vertical wind velocity distributions for RPAS during Flight 30 and cloud radar at RPAS flight altitude (750 m.asl; Fig.17). RPAS measurements are divided into "cloud", and "no cloud" periods. The envelope of each period is plotted based on the minimum and maximum number per bin vertical velocity distributions on a leg-by-leg basis.