# Peer review of "Vertical Wind Velocity Measurements using a 5-hole Probe with Remotely Piloted Aircraft to Study Aerosol-Cloud Interactions"

_Atmospheric Measurement Techniques, 2017_

## Referee Comment (RC1) · Anonymous Referee #1 · 15 Sep 2017

Review of manuscript '3D Wind Vector Measurements using a 5-hole Probe with Remotely Piloted Aircraft' by Radiance Calmer et al.

A) general comments

The manuscript is not well focused. Some statements are questionable. The presented wind-measuring and calibration method is not accurate and in parts wrong. Therefore plenty of confusing corrections were applied to the measurements and lots of discussion / arguing was used to justify missing agreement with other measurements and theory, especially in sections 4 and 5. The results are overall not convincing.

The authors should have a decent look at publications and textbooks that describe the use of flow probes (like a 5-hole probe) aboard small aircraft in detail and then repeat

their raw-data processing.

Sorry for being harsh.

B) specific comments

I am using the following method to name page and line: page/line. E.g. 5/17f means: page 5, line 17 and following lines

* 2/6: 'INS measures six axes': No! It measures 3 linear and 3 circular accelerations or motions.

* 2/14: Having a look in Reuder 2008 shows: SUMO was not able to measure the wind vector. Actually it had no flow sensors aboard at that time. Wind speed and direction were estimated by the drift of the aircraft, instead. An early example of a small and light RPA that actually measured the wind vector was published by Spiess et al in 2007 (already in the list of references).

* 2/19: very unusual and confusing way to cite articles: $M^2AV$-Spiess and MASC-Wildmann! So, the authors names were Spiess (2007) and Wildmann (2014). Aircraft described in these articles were named $M^2AV$ and MASC. Please correct!

* 2/22: Having a look at the sophisticated error analysis of van den Kroonenberg et al (2008), I doubt that Thomas et al (2012) and Baserud (2014) measured the vertical wind 5 times more accurately, the latter with a much simpler measurement system. Please explain how this was possible!

* 2/26 Now measurement of the vertical wind seems to become 20 times more accurate. Explain how this was possible!

* Are these accuracies (from 2/16 to 2/26) all the same by mathematical definition? Do you mean absolute accuracies of the mean vertical wind or resolving turbulent fluctuations? How are the various precisions for the wind velocities in the referenced literature calculated? Are they comparable with each other?

* 2/30 What do you mean with '(4) improved algorithms for wind field estimation from dynamic soaring'?

* 4/19 Indeed, it is possible to use a 5-hole probe with only three differential pressure sensors. But considering publications e.g. by the American Institute of Aeronautics and Astronautics (look for Weiss et al. 1999 to 2002), using 5 differential pressure sensors increases accuracy (measuring the pressure difference of the 5 holes relative to a combined reference), significantly. Since a pressure sensor is light and cheap, can you explain why you decided to use only three?

* 4/10ff (section 2.2) Using a 5-hole probe also requires a proper tubing strategy, see Wildmann et al (2014), in order to avoid signal damping and acoustic tube resonance. Can you please explain how did you take care of tubing issues?

Summing up section 2.2: Why don't you apply well known, state-of-the-art and published (e.g. articles that are already in your list, by van den Kroonenberg and Wildmann et al.) methods for tubing your 5-hole probe? Or in other words: What are the advantages of your unusual method? And can you show that the results are still good enough?

* 4/20 'hole 1 measures the total pressure' - No! And this is a substantial mistake. Since the stagnation point (angle of attack and side slip, \alpha and \beta being not zero) is *somewhere* on the spherical surface of the 5-hole probe, hole 1 does not represent the total pressure! Doing this mistake, the following discussion of wind measurements and accuracies is pointless!

* 4/24f No. An IMU measures accelerations (3 linear and 3 circular). An INS combines an IMU with GPS, usually using a Kalman filter.

* 4/29 'plane'. You mean aircraft or aeroplane.

* 5/1: No! The Eulerian angles are NOT given in the aircraft coordinate system, but in the Earth's coordinate system. This is why people invented INS. And this is important,

if you want to measure the atmospheric wind using an aircraft!

* Entire section 2.3 (page 5): So, you have an INS that delivers the Eulerian angles (roll, pitch, heading). And you have a 5-hole probe that (when properly tubed and calibrated) delivers angle of attack and side slip ($\alpha$ and $\beta$) as well as the total pressure (although you will not have them with the methods described in the manuscript ...). Now you can apply the exact equations, published since the 1970ies, also by Don Lenschow and others, without having to apply any simplification or estimation that only causes worse data. To have a nice and short overview on how to do this, please read (and apply) van den Kroonenberg et al (2008).

* 5/4 Even straight and level flights have varying Eulerian angles. Roll and especially pitch are not zero! The simplified equation (1) for the wind vector possibly holds 1) for heavy and large manned aircraft (talking about airliners) that are less dynamic during flight, and 2) probably only for the estimation of the mean wind vector. But a tiny RPA as used in the presented study is heavily agile. Reading the manuscript I do not see any reason not to use the precise equations (again, see e.g. van den Kroonenberg et al.), or is there any?

* 5/11 $V_e$ etc is confusing. Why don't you use $(V_x, V_y, V_z)$ or $(U,V,W)$ for the ground-speed vector, since you defined your Earth coordinate system in the same paragraph using $(x,y,z)$?

* 6/1 Eq. (2) is not correct. The pressure differences in the denominators have to be divided by the dynamic pressure enhancement, which is the difference between the total and the static pressure. Hole 1 does not deliver the total pressure! See above.

* 6/6 Where does Eq. (3) come from? Please cite literature!

* 6/21: Why do you need a linear relation between $C_{alpha}$/ $C_{beta}$ and alpha/beta?

This is leading to the question how alpha and beta are calculated from the calibration to obtain the wind vector components. Using the system e.g. described in Wildmann

(2014), alpha/beta of up to +-20 degree can be used and a polynomial fit accounts for potential asymmetry of the probe's tip structure.

* Entire section 3.1: There is no need to invent the wheel once more. How to calculate the wind vector and how to calibrate a 5-hole probe is well published and explained. For example in articles in your list of references, see again van den Kroonenberg (2008), there: page 1972f, Eq. (1) to (13).

* 7/7ff: Again, the pressures measured on the dynamic and static pressure holes of the 5-hole probe (in your manuscript delta(P1-P6) and Ps=P6) change with alpha and beta. Moreover other work shows, that the static port of five hole probes fluctuate for inclined inflow.

* 7/9 Why did you calibrate with wind speeds between 12 and 34 m/s? Of course, a single calibration of the 5-hole probe (the entire grid, see Fig. 3 and 4) holds only for one airspeed, i.e. the entire procedure has to be repeated in e.g. 1 m/s steps. This leads to the question, how your autopilot system is controlling the air speed of the RPA?

* 7/13f: What is "triangular motion applied to the pitch axis of the platform"? Why is that performed?

* 7/15: Actually Fig. 5 shows plenty of noise, causing a systematic uncertainty of the vertical-wind measurement of about 0.1 m/s. This means turbulent fluctuations in this order of magnitude cannot be resolved by the presented system. Please make this clear!

* Entire section 3.2: Eventually averaging leads to a mean vertical wind about zero. Having in mind the mean vertical wind should be about zero in the ABL if you average long enough, this is ok if you are only interested in the mean wind. But why is there so much noise? Possibly caused by the electronic pressure transducers?

* 7/28 It is the Gaussian propagation of errors, not the maximum error propagation -

this should be mentioned!

* 8/4, Eq. (6a): please define $a_\alpha$ !

* 8/5, Eq. (6b): please explain the entire equation!

* What is missing in section 3 or 4 is the most simple test to see if the calibration of the system works at least for the mean wind vector: Flying identical legs in opposite direction in a calm atmosphere (e.g. the residual layer, or in an almost neutral stratification under a overcast sky). See also Fig. 6 and Eq. (17) in van den Kroonenberg et al. (2008). Can you show that the heading does not influence the wind measurements?

* Section 4.1: It is very important to check the power spectra of the resulting wind (of course only if the mean-wind check was ok, i.e. identical legs in opposite direction in a calm atmosphere). Spectra show systematic errors as visible in Fig. 7, above 1 Hz (can be the noise level). These are not mentioned in the manuscript - please do so!

However, the mean spectra in Fig. 7 show two critical issues:

1) the Kolmogorov slope is NOT achieved neither with the sonic nor with the RPA data. I do not agree with your text in 9/7. Your spectra have significant different slope. But this can be caused by not having ideal conditions for a locally isotropic turbulent sub-range. More critical (not acceptable) is the following:

2) the spectral power of the RPA data is by a factor of about 5 larger than the spectral power of the sonic. As you mention in 9/8, this was caused by 'the motion of the RPAS'. But if the measured data is governed by the aircraft motion and not by the atmospheric turbulence, any further analysis of turbulence is useless and futile!

* Fig. 8: There is a huge difference between sonic and RPA data around zero vertical wind, please explain!

* Fig. 9: There is a huge difference between mast and RPA data around zero vertical wind. Is there any easy explanation? Section 4.2 is not helpful but confusing.

[Figure]

* 9/13 'This step is needed ...' I do not understand this - the attitude of the aircraft and the 5-hole probe (assuming there is no mounting error) is known from the Eulerian angle delivered by the INS. And the mounting error of the probe to the aircraft is constant.

* Section 4.2: What is the intention of this analysis of the TKE? What shall be learned? Why is it filled with corrections? I am quite sure that these considerations become unnecessary after doing a correct wind-vector calculation (see above).

* 9/30 TKE: From the section 3.3 we know that the uncertainty for the vertical wind is 0.1 m/s. Thus the measurement system causes (possibly by electrical noise) already a standard deviation in this order of magnitude in the data. How large is the uncertainty for the horizontal wind? This is important to know in order to estimate the significance of the presented TKE data.

* 10/2 I doubt that reported TKE deviations between sonic data and other small RPA is increasing faith into the presented method or is explaining any physics. What is your message here?

* Section 5: The differences and shifts in the distributions shown in Fig. 12 to 15 are mainly in the order of magnitude of 0.1 m/s. This is the systematical uncertainty caused by the measurement system and explained before. And Fig 8 shows that the RPA was not able to measure small vertical wind speeds adequately. Considering this, what insights are left?

* 11/28 The abstract says 'are now able to accurately measure ... even in clouds'. But now it is written that water is accumulated within the probe, making it useless. It seems (what could be expected) that 5-hole probes cannot be used in clouds, can they?

* 14/17 'Motions induced ...' Well, this can be expected in case the wind was properly calculated using the correct formulas!

* 14/19 My suspicion: the simplified wind equations (that possibly hold for large and

heavy aircraft) and the faulty calibration caused all the insufficient agreements between the RPA data and other data and theory.

* 14/24 'following Kolmogorov' - actually, not really. See above.

* 14/25 Considering that the Kolmogorov distribution was not measured, I doubt that any isotropy of the turbulent flow can be assumed. Can you prove that the variances of the two horizontal wind components are equal, as written in the text?

* 14/27 This is not surprising. Without a proper heading you cannot measure the wind vector. How accurate was the heading so far?

C) technical corrections

Here are some minor comments:

* Proper use of hyphen! E.g. line 3 on page 2: 'boundary-layer turbulence' would be correct. Also correct would be (in contrast), again in line 3 on page 2: 'aircraft based wind measurements'.

* Language has to be improved, e.g. in line 2 on page 2: 'vectors are an essential parameter' - this doesn't make sense.

* Using the cross 'x' in equations is usually reserved for the vector product, not for normal scalar multiplication.

---

## Referee Comment (RC2) · Anonymous Referee #2 · 5 Oct 2017

The authors present an attempt to conduct atmospheric 3D wind measurements from a light remotely piloted aircraft (<2.5kg) using a combined gps/imu and a 5-hole turbulence probe mounted on the nose of the aircraft. Though this is of significant relevance for the field of atmospheric sciences, e.g. by providing a way to characterize vertical velocities at cloud base level from relatively cheap, almost disposable platforms, I am unfortunately not convinced by the validation of the technique presented here. The manuscript is confusing at times, and will require a careful rewrite to be suitable for publication. I am especially concerned about the interpretation of the results presented in section 4, a critical aspect of this work as it is used as the validation of the RPAS sensor package. The spectral levels, "spikes" and slopes found in the frequency spectra off the three components of the motion compensated winds computed from the

[Figure]

RPAS (and compared against the anemometer ground truth) shown in figure 7 are very concerning and not discuss in depth in the text (I went back to Reineman et al. 2013, there found slight spectral level differences for the lowest wavenumbers, but nothing as significant as what is presented here). I would encourage the authors to go back to the data processing and ensure the algorithm is motion compensating the relative winds computed from the 5-hole probe correctly. Figure 8 further demonstrates the disagreement between the sonic and RPAS wind measurements. I can't really say much about the rest of the manuscript without having resolved the basic measurement concern explained above.

---

## Author Comment (AC1) · 1 Feb 2018

**General comments**

We thank the reviewer for his comments. The manuscript has been restructured in order to address the reviewers' concerns. The main modifications are the calibration of the 5-hole probe, the use of full wind equations instead of simplified equations (Lenschow et al., 1989), and a more detailed analysis of the PSDs to understand the origin of differences between the RPA and mast measurements. We also show that difference in vertical wind distributions does not deeply affect the calculation of cloud droplet number for aerosol-cloud study.

[Figure]

**Specific comments**

Comments from the reviewer appear in italic, response from the authors follows. As the manuscript has been mainly rewritten, we invite the reviewers to directly refer to the updated version of the manuscript for specified sections.

*The manuscript is confusing at times, and will require a careful rewrite to be suitable for publication.*

Response: We have reworked the manuscript to clarify the sections, in particular related to the calibration and the validation of the results with sonic anemometers.

*The spectral levels, "spikes" and slopes found in the frequency spectra off the three components of the motion compensated winds computed from the RPAS (and compared against the anemometer ground truth) shown in figure 7 are very concerning and not discuss in depth in the text.*

Response: The Power Spectral Density (PSD) functions have been updated. The analysis now uses the Welch's method to calculate PSDs. The spikes have been suspected to originate from the INS. We agree that there are still issues to address turbulence or fluxes measurements (particularly calculations of divergence and convergence), as these studies require highly accurate wind measurements. However, for the purpose of aerosol-cloud interaction study, the accuracy of the updraft measurements conducted in the present work has been shown to be sufficient (Section 6).
*I went back to Reineman et al. 2013, there found slight spectral level differences for the lowest wavenumbers, but nothing as significant as what is presented here.*

Response: We thank the reviewer for pointing this statement out. There is actually a good agreement in spectral level for vertical wind in Reineman et al. (2013). However, a difference of level of spectral energy for the vertical wind component is found in Reuder et al. (2016) and Båserud et al. (2016) for the SUMO RPA compared to sonic anemometer. The TKE measurements from the $M^2AV$ in Lampert et al. (2016) is higher than TKE from sonic anemometer on a mast during the afternoon and the night, which also implies higher energy levels of the PSDs. The manuscript has been updated to assess the source of difference in PSDs between the RPA and the mast. We found higher energy in ground speeds provided by the INS for frequencies lower than 0.3 Hz, which influence the wind calculation.

*I would encourage the authors to go back to the data processing and ensure the algorithm is motion compensating the relative winds computed from the 5-hole probe correctly.*

Response: The calibration of the 5-hole probe has been updated, using polynomial instead of linear coefficients for $\alpha$ the angle of attack and $\beta$ the angle of sideslip. Polynomial coefficients from Treaster et al. (1978) method have been determined for static and dynamic pressures, and then used in the calculation of $V_a$ the airspeed. As the RPA operates in the quasi-linear regime of the 5-hole probe (pitch and roll angles < 10 deg), no significant modification of the wind results has been observed with the updated calibration. The full wind equations have been used instead of the simplified equations (Lenschow et al., 1989). A comparison between the wind results has been

conducted to show a difference less than 0.03 $\pm$ 0.05 m/s on straight-and-level legs. However, to avoid any confusion, the full wind equations have been employed in the analysis, except when specified (uncertainty analysis on $w$, section 3.2).

*\*Figure 8 further demonstrates the disagreement between the sonic and RPAS wind measurements.*

Response: We disagree; actually, Figure 8 shows agreement. Uncertainty at low wind speeds related to noise – and the peak at near zero-winds is expected. The limit of detection is $\pm$ 0.1 m/s (also quantified in Figure 5), which is largely sufficient for studying aerosol-cloud interactions. In the manuscript, a section has been added to quantify the influence of the updraft in term of cloud droplet number concentrations.

---

## Author Comment (AC2) · 1 Feb 2018

**General comments**

Both reviewers' responses focus primarily on the method in which the calibration procedure and simplified equations of motion have been presented in this manuscript. One of the main concerns is that we have applied a simplified calibration and calculations of atmospheric winds when full equations have long been published (Lenschow et al., 1989). We are well aware of the Lenschow equations. We are also aware (but did not clearly state in the manuscript) that we are presenting results from a quasi-linear response region of the 5-hole probe, and we had made a conscience choice to structure the manuscript this way. As both reviewers asked for

clarification regarding the calibrations and equations of motion, we have modified the manuscript with citations for the full set of equations (Lenschow et al. (1989); Boiffier (1998); Kroonenberg et al. (2008)) and simply mention in the manuscript that when operating within a linear regime (ca. $\pm$ 10 degrees), and that simplified equations may be used for evaluating results and assessing uncertainties of atmospheric wind measurements. We also show in the responses here and in the manuscript, that the main results related to the power spectra, calculation of turbulent kinetic energy and the atmospheric winds do not change for our flight conditions, even after accounting for nonlinearity in the calibration coefficients and the full set of Lenschow equations.

We also highlight that this manuscript presents results using a commercially-available probe with custom electronics, and an INS and RPA that have not been previously reported in the literature. Consequently, we dedicate a section of the manuscript to the calibration/validation of the 5-hole probe in order to compare with other multi-hole probes in the literature. As the 5-hole probe and associated sensors, the INS, and the RPA comprise the measurement system, all components should be validated together as a system. We present the results of turbulent kinetic energy (TKE) in order to compare our results to other such RPA measurements (i.e., Båserud et al. (2016); Lampert et al. (2016); Canut et al. (2016)). We are not aware of any direct comparisons of vertical wind velocity between ground-based remote-sensing instruments (e.g., lidar, cloud radar) and a multi-hole probe mounted on a RPA. The vertical winds derived from the 5-hole probe have been implemented in aerosol-cloud parcel models (e.g., Sanchez et al. (2017)), and measurements of updraft velocity to within 0.1 m s$^{-1}$ are largely sufficient for conducting aerosol-cloud closure studies. As the focus of this work is obtaining updraft measurements, we have updated the title of the manuscript to reflect this. Finally, we included a new section that describes the sensitivity of vertical velocity distributions on resulting cloud droplet number concentrations.

**Specific comments**

The same method is used to name page and line number : page/line. Comments from the reviewer appear in italic, response from the authors follows as well as the updated section of the manuscript. However, as the manuscript has undergone significant changes, we invite the reviewers to directly refer to the updated version of the manuscript for specified sections.

RPAS (Remotely Piloted Aircraft System) is used to name the global system, including aircraft + ground control station, and RPA (Remotely Piloted Aircraft) only refers to the aircraft.

*2/6: 'INS measures six axes': No! It measures 3 linear and 3 circular accelerations or motions.*

Response: We are clearly referring to the six degrees of freedom of the INS (as noted by the reviewer; three linear and three rotational). These six degrees of freedom are often denoted as six-axis sensors by the manufacturers (i.e., TDK, Bosch) and peer-reviewed literature in robotics. Nonetheless, we will remove the sentence in the manuscript to avoid any confusion.

In the manuscript at 2/6 : "INS measure linear and rotational motion of the aircraft (or unmanned aerial system) and are used to back out wind vectors in the Earth's coordinate system."

*2/14: Having a look in Reuder 2008 shows: SUMO was not able to measure the wind vector. Actually it had no flow sensors aboard at that time. Wind speed and direction were estimated by the drift of the aircraft, instead. An early example of a small and light RPA that actually measured the wind vector was published by Spiess et al in 2007 (already in the list of references).*

Response : We thank the reviewer for pointing this out and we have updated the manuscript to cite Spiess et al. (2007). We also clarify that Reuder et al. (2008) do not calculate wind speed and direction by the drift of the RPA, rather a vector calculation using minimum and maximum ground speeds and respective orientations along a circular path around a fixed GPS coordinate.

In the manuscript at 2/5 sentence now reads : "to ultra-light unmanned aerial systems (i.e., $M^2AV$; Spiess et al (2007))." and at 2/14 : "to a 600 g SUMO (Reuder et al., 2012)"

*2/19: very unusual and confusing way to cite articles: M2AV-Spiess and MASCWild-mann! So, the authors names were Spiess (2007) and Wildmann (2014). Aircraft described in these articles were named M2AV and MASC. Please correct!*

Response : Description of RPAS has been re-formulated as the reviewer suggested.

*\* 2/22: Having a look at the sophisticated error analysis of van den Kroonenberg et al (2008), I doubt that Thomas et al (2012) and Baserud (2014) measured the vertical*

*wind 5 times more accurately, the latter with a much simplier measurement system. Please explain how this was possible!*

Response : The uncertainty associated with the wind measurement is a combination of the errors from the probe and the INS.Thomas et al. (2012) and Reineman et al. (2013) use much more precise INS which allow them to report smaller uncertainties. Even if Båserud et al. (2016) and Thomas et al. (2012) both implemented an Aeroprobe instrument (membrane-based sensors), the uncertainties reported in their respective papers are not comparable. Båserud et al. (2016) did not include the INS error in the estimation of the wind uncertainty. We have clarified this in the manuscript. Updates on the wind uncertainty from RPAs and the method to obtain it are provided in the manuscript (see next two comments for updated text).

*\* 2/26 Now measurement of the vertical wind seems to become 20 times more accurate. Explain how this was possible!*

Response: The increase in accuracy is largely due to the INS model selected for wind measurements. The following uncertainties are provided by Reineman et al. (2013) for the dGPS/IMU on board of the Manta RPA, 0.007/0.007/0.011 degrees for roll/pitch/heading, respectively, and 0.01 m/s for the INS vertical velocity. The uncertainties for the INS on the $M^2AV$ (Kroonenberg et al., 2008), are higher, 0.54/0.71/1.22 degrees for roll/pitch/heading, respectively, and 0.58 m/s for the INS vertical velocity. All these parameters present in Reineman et al. (2013) composed a system similar to the BAT probe mounted on a small manned aircraft (Garman et al. (2006), Crawford et al. (1992)).

*\*Are these accuracies (from 2/16 to 2/26) all the same by mathematical definition? Do you mean absolute accuracies of the mean vertical wind or resolving turbulent fluctuations? How are the various precisions for the wind velocities in the referenced literature calculated? Are they comparable with each other?*

Response : As the reviewer suggests, the reported uncertainties are not necessarily consistent by the same mathematical definition and a complete analysis is beyond the scope of this paper. Indeed, this is an important point, which we have attempted to clarify this in the manuscript.

In the manuscript 2/14 :"A wide range of remotely piloted aircraft (RPA)[1] has been used to measure atmospheric winds, from a 600 g SUMO (Reuder et al., 2012) to a 30 kg Manta (Thomas et al., 2012). In particular, a multi-hole probe paired with an INS has been the main mechanism for obtaining vertical winds in fixed-wing RPA. Ultimately, the combination of the multi-hole probe, pressure sensors, and the INS dictates the precision of atmospheric wind measurements. The following accuracies for vertical wind measurements $w$ were reported in the literature for different RPA platforms; they were obtained by different methods, which provided either 1-sigma uncertainty or systematic error analysis associated with a specific pair of probe/INS. In Kroonenberg et al. (2008), a custom 5-hole probe on the $M^2AV$, implemented with a GPS-MEMS-IMU was reported with an accuracy for $w$ within $\pm$ 0.5 m s$^{-1}$. The accuracy was based on a systematic error estimation using characteristic flight parameters with a reference state of $w$ = 1 m s$^{-1}$. The uncertainty in $w$ reported for the SUMO (Reuder et al., 2016) is $\pm$ 0.1 m s$^{-1}$ as given by the manufacturer (Aeroprobe Corporation); however, the impact of the INS was not included in their analysis. In Thomas et al. (2012), the Manta RPA was also equipped with a commercial Aeroprobe and a C-Migits-III INS to obtain a minimum resolvable $w$ of 0.17 m s$^{-1}$ (1-sigma). The uncertainty analysis was based on a Gaus-
* * *
[1]RPA refers to the aircraft, as RPAS represents the airframe and the ground control station

sian error propagation described in Garman et al. (2006). The Manta and ScanEagle RPAs described in Reineman et al. (2013) achieved precise wind measurements with a custom 9-hole probe and NovAtel INS with reported uncertainties for $w$ within $\pm$ 0.021 m s$^{-1}$. Their uncertainty was obtained from a Monte-Carlo simulation, and was also consistent with reverse-heading maneuvers. The higher precision reported in the latter study Reineman et al. (2013) is related to probe design and the high precision of the INS. The vertical wind measurements in Reineman et al. (2013) have a similar performance as reported with the BAT probe ("Best Air Turbulence Probe") on a small piloted aircraft (Garman et al., 2006)."

*\* 2/30 What do you mean with '(4) improved algorithms for wind field estimation from dynamic soaring'?*

Response : Elston et al. (2015) reviewed RPAS in atmospheric science and included methods for wind field estimation based on dynamic soaring. We have removed this point as it does not relate directly to our manuscript.

*\* 4/19 Indeed, it is possible to use a 5-hole probe with only three differential pressure sensors. But considering publications e.g. by the American Institute of Aeronautics and Astronautics (look for Weiss et al. 1999 to 2002), using 5 differential pressure sensors increases accuracy (measuring the pressure difference of the 5 holes relative to a combined reference), significantly. Since a pressure sensor is light and cheap, can you explain why you decided to use only three?*

[Figure]

Response : There exist multiple methods for configuring the pressure sensors as described in peer-review literature. Wildmann et al. (2014) present two possibilities implemented on the $M^2AV$ (fig.4) and on the MASC (fig.5). Reineman et al. (2013) use a similar differential pressure configuration as the present study. The five pressure sensor system used by Reineman et al. (2013) measured two additional axes (9 holes with a 45 degree component between vertical and horizontal axes) to improve performance of their probe. Three differential pressure measurements method is also proposed by the Aeroprobe Corporation (calibration files provided by the manufacturer) as well as the 5 hole-probe implemented by Thomas et al. (2012) on the Manta and Reuder et al. (2016) on the SUMO. Adding five differential pressure sensors is only a valid argument if the sum of the measurement errors of the individual sensors is less than a single sensor (which would not be the case for the sensors that we are using). In addition, using five differential pressure sensors requires additional resources on the data acquisition system (which we did not have), and more volume in the payload bay (an extra printed circuit board).

*\* 4/10ff (section 2.2) Using a 5-hole probe also requires a proper tubing strategy, see Wildmann et al (2014), in order to avoid signal damping and acoustic tube resonance. Can you please explain how did you take care of tubing issues?*

Response : Indeed, we are aware of the tubing issues extensively studied in Wildmann et al. (2014). In the present study, the tubing length is less than 15 cm, and the inner diameter is 0.1 mm, which are dimensions similar to Wildmann et al. (2014); and the frequency of our measurements is less than that verified by Wildmann et al. (2014). We have specified this in the manuscript.

*\*Summing up section 2.2: Why don't you apply well known, state-of-the-art and published (e.g. articles that are already in your list, by van den Kroonenberg and Wildmann et al.) methods for tubing your 5-hole probe? Or in other words: What are the advantages of your unusual method? And can you show that the results are still good enough?*

Response : As noted in the opening statements of our response, we were operating in the nearly linear response regime of the probe. In addition, we used a configuration proposed by the manufacturer of the probe and methods that are also described in a thesis by Truong (2011). None of the results in this manuscript (PSD, TKE, updraft) significantly change when using the full expressions reported in Kroonenberg et al. (2008) and Wildmann et al. (2014) because we operated the probe within a quasi-linear regime (at small pitch and roll angles). Nonetheless, we recognize that a thesis and manufacturer's procedures are not as easily accessible as peer-reviewed literature. Consequently, we have updated Section 2.2 with citations to Wildmann et al. (2014), whose procedures are similar to those used in our manuscript. The calibration of the probe in the wind tunnel (Section 3), as well as the Cal/Val with the sonic anemometer (Section 4) and cloud radar (Section 5) confirm the probe's performance.

*4/20 'hole 1 measures the total pressure' - No! And this is a substantial mistake. Since the stagnation point (angle of attack and side slip, nalpha and nbeta being not zero) is \*somewhere\* on the spherical surface of the 5-hole probe, hole 1 does not represent the total pressure! Doing this mistake, the following discussion of wind measurements and accuracies is pointless!*

Response : P1 is referred as the total pressure (Truong, 2011) or as the pressure

at the stagnation point (Reineman et al., 2013). To be consistent with the reviewer's comments, P1 is denoted as the pressure at the stagnation point in the manuscript, and Fig.1.a is updated. An update of the calibration method has been provided in responses of comments about Section 3.1.

*\* 4/24f No. An IMU measures accelerations (3 linear and 3 circular). An INS combines an IMU with GPS, usually using a Kalman filter.*

Response : In the literature INS or IMU are sometimes used incorrectly for the same meaning. The manuscript has been updated to use INS to refer to the system that combines an IMU with GPS.

*\* 4/29 'plane'. You mean aircraft or aeroplane*

Response : To be consistent with the text, the terms "RPA" or "RPAS" are used.

In the manuscript 5/5, "the measured motion of the RPA (given by the INS)".

*\*5/1: No! The Eulerian angles are NOT given in the aircraft coordinate system, but in the Earth's coordinate system. This is why people invented INS. And this is important,*

*if you want to measure the atmospheric wind using an aircraft!*

Response : Indeed, the Eulerian angles are neither given in the aircraft coordinate system nor in the Earth's coordinate system. They represent the orientation of the aircraft with respect to the Earth's coordinate system and build the transformation matrix to convert angles and velocity from one coordinate system to another. We have clarified this point in the manuscript.

In the manuscript 5/6: "The angle of attack, the angle of sideslip, and the airspeed $V_a$ are measured by the 5-hole probe in the probe coordinate system and then transform to the RPA coordinate system; while the attitude angles from the INS provide the transformation of $\alpha$, $\beta$ and $V_a$ from the RPA coordinate system to the Earth's coordinate system. Lenschow equations (Lenschow and Spyers-Duran, 1989) are therefore followed to calculate the wind vector in the Earth's coordinate system. "

*\* Entire section 2.3 (page 5): So, you have an INS that delivers the Eulerian angles (roll, pitch, heading). And you have a 5-hole probe that (when properly tubed and calibrated) delivers angle of attack and side slip (nalpha and nbeta) as well as the total pressure (although you will not have them with the methods described in the manuscript ...). Now you can apply the exact equations, published since the 1970ies, also by Don Lenschow and others, without having to apply any simplification or estimation that only causes worse data. To have a nice and short overview on how to do this, please read (and apply) van den Kroonenberg et al (2008).*

Response : As stated previously, we are well aware of the exact equations, as well

as the limits of the simplified equations. Consequently, we specifically designed the experiment to follow straight-and-level flight legs. Pitch and roll were almost always less than $\pm$ 10 degrees, which is in a quasi-linear response regime of the multi-hole probe. A comparison of the three wind components, $u, v, w$, calculated with the complete Lenschow equations and with the simplified equations has been conducted on straight-and-level legs for the five flights at CRA, Lannemezan. The mean value of the difference in the calculated winds between complete and full equations is less than $0.03 \pm 0.05$ m/s. The difference is less than the uncertainty based on the wind tunnel calibration.

*\* 5/4 Even straight and level flights have varying Eulerian angles. Roll and especially pitch are not zero! The simplified equation (1) for the wind vector possibly holds 1) for heavy and large manned aircraft (talking about airliners) that are less dynamic during flight, and 2) probably only for the estimation of the mean wind vector. But a tiny RPA as used in the presented study is heavily agile. Reading the manuscript I do not see any reason not to use the precise equations (again, see e.g. van den Kroonenberg et al.), or is there any?*

Response : The reviewer is rightfully concerned about the stability of the RPA, and maximum pitch and roll for the five flights in Lannemezan have been investigated. The roll angle exceeds 10 degrees less than 1% of the time, except for Flight 4 (unstable flight close to stall speed) where roll exceed 10 degrees up to 2 % of the time. The pitch angle never exceeded 10 degrees except when it approached stall speed (Flight 4) – even then, pitch angles did not exceed 10 degrees during the straight-and-level legs more than 0.1 % of the time.

*\* 5/11 $V_e$ etc is confusing. Why don't you use $(V_x, V_y, V_z)$ or $(U, V, W)$ for the ground-speed vector, since you defined your Earth coordinate system in the same paragraph using (x,y,z)?*

Response : We agree. In the manuscript $V_p$ for the RPA vertical speed has been replaced by $V_z$.

*\* 6/1 Eq. (2) is not correct. The pressure differences in the denominators have to be divided by the dynamic pressure enhancement, which is the difference between the total and the static pressure. Hole 1 does not deliver the total pressure! See above.*

Response : As stated previously, the method described in the manuscript follows the Aeroprobe corporation calibration file. However, to be consistent with the peer-reviewed literature, the calibration in the manuscript has been updated from the Aeroprobe corporation method to the Treaster et al. (1978) method, also used in Wildmann et al. (2014). A polynomial fit has been obtained from $C_\alpha$ and $C_\beta$ to calculate $\alpha$ and $\beta$. The difference between the previous calibration method and the updated results of the angles is small for angles within ca. $\pm$ 10 degrees (Fig.1 and Fig.2).

*\* 6/6 Where does Eq. (3) come from? Please cite literature!*

Response : Equation (3) comes from the Aeroprobe calibration file, and can also be found in the literature (Truong, 2011). As noted earlier, we have updated the equations

and cited Wildmann et al. (2014).

*\* 6/21: Why do you need a linear relation between $C_\alpha$/ $C_\beta$ and alpha/beta? This is leading to the question how alpha and beta are calculated from the calibration to obtain the wind vector components. Using the system e.g. described in Wildmann (2014), alpha/beta of up to +-20 degree can be used and a polynomial fit accounts for potential asymmetry of the probe's tip structure.*

Response : Figure 4 in the manuscript has been updated with the new $C_\alpha$ and $C_\beta$.

*\* Entire section 3.1: There is no need to invent the wheel once more. How to calculate the wind vector and how to calibrate a 5-hole probe is well published and explained. For example in articles in your list of references, see again van den Kroonenberg (2008), there: page 1972f, Eq. (1) to (13).*

Response: As mentioned, we are using a commercially available 5-hole probe (Aeroprobe) with custom electronics. We are well aware of assumptions that have been made; however, due to our configuration of pressure sensors and tubing, we have revised our calibration using the Treaster et al. (1978) method (used in Wildmann et al. (2014)), rather than the method based on Bohn et al. (1975) (used in van den Kroonenberg et al. (2008)).

*\*7/7ff: Again, the pressures measured on the dynamic and static pressure holes of*
*the 5-hole probe (in your manuscript delta(P1-P6) and Ps=P6) change with alpha and beta. Moreover other work shows, that the static port of five hole probes fluctuate for inclined inflow.*

Response : The coefficients $C_q$ and $C_s$ from the Treaster et al. (1978), and Wildmann et al. (2014), are calculated to obtain the airspeed as a function of $\alpha$ and $\beta$ for our probe (Fig.5). The airspeed is then calculated with $P_s$ and $P_q$ (Table A1 in Wildmann et al. (2014)) from Anderson (2001) with speed of sound and Mach number, or from Wildmann et al. (2014) with the Poisson number.

*\* 7/9 Why did you calibrate with wind speeds between 12 and 34 m/s? Of course, a single calibration of the 5-hole probe (the entire grid, see Fig. 3 and 4) holds only for one airspeed, i.e. the entire procedure has to be repeated in e.g. 1 m/s steps. This leads to the question, how your autopilot system is controlling the air speed of the RPA?*

Response : Variation of the wind tunnel airspeed from 12 to 34 m s$^{-1}$ simply verifies the calibration of the probe in the range of RPA operating airspeed and the instrument uncertainties. The difference between the polynomial fits at distinct airspeeds is relatively small and leads to less than a 0.6 degree difference between $\alpha$ or $\beta$ (Fig.1). The RPAS controls airspeed by pitch adjustment.

*\* 7/13f: What is "triangular motion applied to the pitch axis of the platform"? Why is that performed?*

Response : Triangular motion is applied to validate the calibration in a controlled environment and verify the response of the INS. The figure (Fig.5 in the manuscript) shows a larger uncertainty related to faster motions induced on the probe and INS. We suspect the increased uncertainty is related to an induced lag in the filtering process within the INS.

*\* 7/15: Actually Fig. 5 shows plenty of noise, causing a systematic uncertainty of the vertical-wind measurement of about 0.1 m/s. This means turbulent fluctuations in this order of magnitude cannot be resolved by the presented system. Please make this clear!*

Response : We clarify in the text that turbulent fluctuations on the scale of 0.1 m s$^{-1}$ cannot be resolved. The figure showing the time series of the triangular motions have been removed from the revised manuscript to avoid any confusion.

*\* Entire section 3.2: Eventually averaging leads to a mean vertical wind about zero. Having in mind the mean vertical wind should be about zero in the ABL if you average long enough, this is ok if you are only interested in the mean wind. But why is there so much noise? Possibly caused by the electronic pressure transducers?*

Response : The averaging is conducting over time scales of the flight, which are sufficiently large. As stated in the response 7/13, the high frequency motions of
the two-axis platform increase the noise in $w$ calculation. The uncertainty remains acceptable as the standard deviation of $w$ is less than 0.1 m/s (Fig.5 in the manuscript).

*\* 7/28 It is the Gaussian propagation of errors, not the maximum error propagation - this should be mentioned!*

Response : The manuscript has been updated and the section 3.2 is named Gaussian error propagation on vertical wind velocity.

*\* 8/4, Eq. (6a): please define $a_{alpha}$ !*

Response : We thank the reviewer for catching this oversight and have defined $a_\alpha$. (see next comment for the updated text)

*\* 8/5, Eq. (6b): please explain the entire equation!*

Response : The section 3.2 has been rewritten to provide more details and clearer explanation about the conducted uncertainty analysis. Please, refer to 6/29 in the manuscript.

*\* What is missing in section 3 or 4 is the most simple test to see if the calibration of the system works at least for the mean wind vector: Flying identical legs in opposite direction in a calm atmosphere (e.g. the residual layer, or in an almost neutral stratification under a overcast sky). See also Fig. 6 and Eq. (17) in van den Kroonenberg et al. (2008). Can you show that the heading does not influence the wind measurements?*

Response : Actually, a drift in the heading does influence the horizontal wind calculation. We have flown identical legs in the opposite direction to specifically show that the wind vectors are similar (within the instrument uncertainty). We show a time series of such results here, and state that such verifications have been done in the manuscript. The influence of the drift in the heading is clearly visible in the calculation of the $u$-component of wind, as $u$ increases along each leg (in blue in Fig.5 in the present document). The uncertainty in horizontal winds is 1.1 m/s.

*\* Section 4.1: It is very important to check the power spectra of the resulting wind (of course only if the mean-wind check was ok, i.e. identical legs in opposite direction in a calm atmosphere). Spectra show systematic errors as visible in Fig. 7, above 1 Hz (can be the noise level). These are not mentioned in the manuscript - please do so!*

Response : We are aware of the error visible on the $v$-component above 1 Hz for Flight 1, 2 and 3 (green in Fig.6b in the revised manuscript). However, the error has been removed by reconfigurating the INS as shown in the PSD of Flight 5 (blue in Fig.6b in the revised manuscript) .

[Figure]

*However, the mean spectra in Fig. 7 show two critical issues: 1) the Kolmogorov slope is NOT achieved neither with the sonic nor with the RPA data. I do not agree with your text in 9/7. Your spectra have significant different slope. But this can be caused by not having ideal conditions for a locally isotropic turbulent sub-range.*

Response : The Welch method has been used to plot the PSDs. The slope of the wind PSDs from the RPA data follows the -5/3 line.

*More critical (not acceptable) is the following: 2) the spectral power of the RPA data is by a factor of about 5 larger than the spectral power of the sonic. As you mention in 9/8, this was caused by 'the motion of the RPAS'. But if the measured data is governed by the aircraft motion and not by the atmospheric turbulence, any further analysis of turbulence is useless and futile!*

Response : This is a known issue, Reuder et al. (2016) report a SUMO overall energy level that is also higher than that of the sonic anemometer. The reason for the systematic difference has been investigated for the present work, using the decomposition of the wind equations to better understand the influence of each terms : airspeed, ground speeds, heading. Sampling frequencies and new/old calibrations have also been compared without showing any differences in the PSDs. The ground speeds obtained from the INS ($V_x$, $V_y$, $V_z$) present systematic higher energy level for frequencies less than 0.3 Hz. More efforts are going to be invested in selecting the parameters for the extended Kalman filter of the INS, and a comparison with another INS model is going to be conducted to select the best instrument. We recognize that our current measurements are not be suitable for convergence/divergence calculations, however, we demonstrate that for aerosol-cloud interactions, the accuracy achieved with our

updraft measurements is valid for estimation of cloud droplet number (Section 6).

*\* Fig. 8: There is a huge difference between sonic and RPA data around zero vertical wind, please explain!*

Response : The difference around zero vertical wind is simply due to instrument uncertainty reported in Figure 5 in the manuscript. The lower limit in vertical wind measurements is ca. 0.1 m/s.

*\* Fig. 9: There is a huge difference between mast and RPA data around zero vertical wind. Is there any easy explanation? Section 4.2 is not helpful but confusing.*

Response : We have restructured Section 4 to clarify.

*\* 9/13 'This step is needed ...' I do not understand this - the attitude of the aircraft and the 5-hole probe (assuming there is no mounting error) is known from the Eulerian angle delivered by the INS. And the mounting error of the probe to the aircraft is constant.*

Response : Throughout the course of a field campaign, we detected relatively small

differences in the performance of the probe related to its alignment. Exposure to the sun altered the wing profile and control surfaces, coupled with repeated net landings, the alignment of the probe relative to the airstream did change – consequently we verify/correct for these changes for each flight.

*\* Section 4.2: What is the intention of this analysis of the TKE? What shall be learned? Why is it filled with corrections? I am quite sure that these considerations become unnecessary after doing a correct wind-vector calculation (see above).*

Response : The analysis of TKE was selected simply to compare the measurements from our RPA to those published from other platforms. Having observed non-isotropy in the RPA's transversal wind component, we then seek to improve TKE calculations. Such corrections are similar to the method proposed in Lampert et al. (2016) to obtain TKE from the M$^2$AV platform during the BLLAST campaign.

*\* 9/30 TKE: From the section 3.3 we know that the uncertainty for the vertical wind is 0.1 m/s. Thus the measurement system causes (possibly by electrical noise) already a standard deviation in this order of magnitude in the data. How large is the uncertainty for the horizontal wind? This is important to know in order to estimate the significance of the presented TKE data.*

Response : The uncertainty in horizontal wind results primarily from drift in the INS heading and is 1.1 m/s (based on out-and-back flights along straight-and-level legs).

Note this uncertainty is largely driven by the uncertainty related to the transversal wind component.

*\* 10/2 I doubt that reported TKE deviations between sonic data and other small RPA is increasing faith into the presented method or is explaining any physics. What is your message here?*

Response : To our knowledge, there is no direct comparison of updraft measurements between a RPA platform and ground-based remote-sensing instruments; consequently, we calculate TKE to compare/validate our wind measurements with previous peer-reviewed results.

*\* Section 5: The differences and shifts in the distributions shown in Fig. 12 to 15 are mainly in the order of magnitude of 0.1 m/s. This is the systematical uncertainty caused by the measurement system and explained before. And Fig 8 shows that the RPA was not able to measure small vertical wind speeds adequately. Considering this, what insights are left?*

Response : We attempt to clarify in the manuscript, 1) that the lower limit of measuring vertical wind is 0.1 m/s which is largely sufficient for aerosol-cloud interaction studies, and 2) measurements of the wind-RPA and the cloud radar do not sample the same air mass, so agreement within 0.1 m/s implies that both observing systems (cloud radar and RPA) measure the same state of the boundary layer. The insight of this work lies

in the ability of the RPA to identify different states of the boundary layer (in/out of cloud, over water/land), and to provide a range of vertical wind velocities near cloud base to accurately determine the number of aerosol particles that activate into cloud droplets.

*\* 11/28 The abstract says 'are now able to accurately measure ... even in clouds'. But now it is written that water is accumulated within the probe, making it useless. It seems (what could be expected) that 5-hole probes cannot be used in clouds, can they?*

Response : Yes, in fact, the 5-hole probe can be used in clouds – we clearly show this in the comparison with the cloud radar at Mace Head. We agree that the probe can be improved, and we are addressing this issue.

*\* 14/17 'Motions induced ...' Well, this can be expected in case the wind was properly calculated using the correct formulas!*

Response : We have addressed this issue in response 5/4.

*\* 14/19 My suspicion: the simplified wind equations (that possibly hold for large and heavy aircraft) and the faulty calibration caused all the insufficient agreements between the RPA data and other data and theory.*

[Figure]

Response : As it has been addressed in the previous responses, the linear approximation for the calibration coefficients and the simplified wind equations do not affect the results as we operated the RPA within the quasi-linear response regime of the probe, on straight-and-level legs.

*\* 14/24 'following Kolmogorov' - actually, not really. See above.*

Response : We have addressed this issue in response Section 4.1, in Fig.6.

*\*14/25 Considering that the Kolmogorov distribution was not measured, I doubt that any isotropy of the turbulent flow can be assumed. Can you prove that the variances of the two horizontal wind components are equal, as written in the text?*

Response : Perhaps it is not clear, but we invite the reviewer to take a closer look at Fig.9 in the manuscript, which compares the variances of the horizontal wind components from the sonic anemometers at different heights. The results show that the isotropy assumption is a better approximation at 60 m.agl compared to 30 m.agl. We show these results to support the arguments for improving the calculation of TKE and because data from the mast at 60 m.agl was not always available for comparison to the RPA.

*\* 14/27 This is not surprising. Without a proper heading you cannot measure the wind vector. How accurate was the heading so far?*

Response : Of course, drift in heading is a known issue for INS. The calibration of the magnetometer helps; however, its time response is relatively slow. Reineman et al. (2013) and Thomas et al. (2012) address this issue with state-of-the-art INS that are much more precise (as well as expensive and heavy). INS technology continues to evolve, and as stated in the conclusions, we are addressing these issues by incorporating an INS with a differential GPS system.

**Technical corrections**

*\* Proper use of hyphen! E.g. line 3 on page 2: 'boundary-layer turbulence' would be correct. Also correct would be (in contrast), again in line 3 on page 2: 'aircraft based wind measurements'.*

Response: We updated the text.

*\* Language has to be improved, e.g. in line 2 on page 2: 'vectors are an essential parameter' - this doesn't make sense.*

Response : The manuscript has been updated.

In the manuscript 2/1 : "Vertical wind is a key parameter for understanding aerosol-cloud interactions."

*\* Using the cross 'x' in equations is usually reserved for the vector product, not for normal scalar multiplication.*

Response : The cross 'x' has been removed from equations to avoid any confusion.

[Figure]

[Figure]

**Fig. 1.** Comparison of the previous calibration with the Aeroprobe calibration method and the updated calibration with method from Treaster et al. (1978) for Ca

[Figure]

**Fig. 2.** Comparison of the previous calibration with the Aeroprobe calibration method and the updated calibration with method from Treaster et al. (1978) for Cb

[Figure]

- $\alpha = 0\ deg$ (blue)
- $\alpha = 5\ deg$ (magenta)
- $\alpha = 10\ deg$ (green)
- $\alpha = 15\ deg$ (black)
- $\alpha = -5\ deg$ (red)
- $\alpha = -10\ deg$ (yellow)
- $\alpha = -15\ deg$ (cyan)

**Fig. 3.** Dynamic pressure coefficient function of angle of sideslip. Crosses correspond to measurement points, and curve to polynomial best fit. The wind tunnel velocity is set up to 15 m/s.

[Figure]

**Fig. 4.** Static pressure coefficient function of angle of sideslip. Crosses correspond to measurement points, and curve to polynomial best fit. The wind tunnel velocity is set up to 15 m/s.

[Figure]

**Fig. 5.** Flight 5 at CRA, Lannemezan, time series of the East-West wind (u-component) measured from the wind-RPA. Straight-and-level legs are represented in blue.

---

## Author Response (AR2)

**Interactive comment on "Vertical Wind Velocity Measurements using a 5-hole Probe with Remotely Piloted Aircraft to Study Aerosol-Cloud Interactions"**

Response to referee #1

**Minor revisions**

We thank the reviewer for his comments.

Comments from the reviewer appear in italic, response from the authors follows.

*For the introduction, I have some suggestions, all published in high-ranked journals:*

*\* another nice example for atmospheric boundary layer studies of turbulent fluxes using RPAS is*

*Wildmann N., Rau G.A., and Bange J., 2015: Observations in the early morning boundary layer transition with small RPA. Boundary-Layer Meteorol., 157, 345?373.*

*\* and for precise ABL wind vectors using RPAS:*

*Wildmann N., Bernard S., and Bange J., 2017: Measuring the local wind field at an escarpment using small remotely-piloted aircraft. Renewable Energy, 103, 613?619.*

*\* and for aerosol in the ABL using RPAS:*

*Platis A., Altstädter B., Wehner B., Wildmann N., Lampert A., Hermann M., Birmilli W., and Bange J., 2016: An observational case study on the influence of atmospheric boundary-layer dynamics on new particle formation. Boundary-Layer Meteorol., 158, 67-92.*

Response: We thank the reviewer for bringing to our attention these recent publications. We have cited an article to reinforce the main points in the introduction.

In the manuscript 3/14 : A study of new particle formation in the atmospheric boundary layer has been conducted by Platis et al. (2015), using the MASC and ALADINA RPAs. Vertical profiles

during the short morning transition between shallow convective to mixed boundary layer highlight the important role of turbulence in new particle formation processes.

Vertical profiles during the short morning transition between shallow convective to mixed boundary layer highlight the important role of turbulence in new particle formation processes.

*4/23: "hole 1 measures the pressure at the stagnation point of the tip" - still, this is not correct, as explained in my first review. However, at small angles of attack and sideslip, the error caused by this misunderstanding might not be large. I am frustrated to see that this mistake was done in other publications too, obviously. Correct: "hole 1 gives an estimate of the pressure at the stagnation point of the tip"*

Response: The text has been updated with the reviewer's comments.

In the manuscript 4/23 : "Figure 1a illustrates the probe schematic: hole 1 gives an estimate of the pressure at the stagnation point of the tip;"

*Sections 2.3 and 3: And I still do not see the point in using simplified equations that cause uncertainties regarding the measured wind vectors while the precise equations are well known and can by applied easily. The analysis of errors caused by 10 degrees of angles of attack and sideslip is more costly than the application of the correct equations. Anyway, this issue is now disarmed by citing the appropriate literature.*

Response: The full set of wind equations are used for all wind calculations throughout the manuscript. To clarify this point, text has been added in the manuscript. The simplified equation of vertical wind $w$ is used only to facilitate the presentation of the error propagation of vertical winds.

In the manuscript 5/9 : "The full set of wind equations from Lenschow and Spyers-Duran (1989) are followed throughout this study to derive the atmospheric wind vectors in the Earth's coordinate system."

*5/20 and Tab 1: SI units instead of 'mbar' would be contemporary*

Response: We agree; the unit 'mbar' has been changed to 'Pa' in the text and in Table 1

In the manuscript : 5/20 "The calibration of the 5-hole probe is a two-step process — first calibrating the differential pressure sensors (Pa), then associating the differential pressures in Pa to angles ($\alpha$ and $\beta$; degrees) and airspeed ($V_a$; m s$^{-1}$)."

*The discussion of the power spectra (section 4.1) and the TKE (4.2) is still weak. The argument that other publications show similar deviations e.g. to sonic anemometers is not satisfactory, since no reason was found for this systematic discrepancy (?be further investigated in the future?). However, the reason could also be faulty sonic measurements of turbulent fluctuations.*

Response: We have indeed suspected a malfunction of the sonic anemometer for at least one case (Flight 5). The topography and also the tower itself may affect the sonic anemometer measurements. These points establish a sampling strategy in an extended experiment that is beyond the scope of the present study (for example, with a tethered balloon or even a piloted research aircraft).

*Fig. 12, 15 and 18: The meaning of captions 'Radar flight altitude' etc (explanations to the four data curves) are not clear to me.*

Response: The legends 'Radar flight altitude', 'Radar flight time', 'Radar cloud top' and 'RPA flight 26' refer to the time series and altitudes defined in the associated figures from the cloud radar (Figure 11). The legends of Fig. 12, 15 and 18 have been updated to clarify the meaning of captions.

In the manuscript 27/Figure 12. "Comparison of vertical wind velocity distributions in a lightly precipitating stratocumulus deck between RPA (1160 m.asl), cloud radar at RPA altitude (1160 m.asl), and cloud radar at cloud top (1360 m.asl). "Radar Flight altitude" corresponds to 4 hours of cloud radar measurements at the same altitude as the RPA flight. "Radar flight time" corresponds to the cloud radar measurements during the RPA flight period at the same altitude as the RPA. "Radar cloud top" corresponds to cloud radar measurements near the cloud top, which is in the non-precipitating part of the cloud. "RPA flight 26" corresponds to RPA vertical wind measurements during Flight 26. Time periods and altitudes are identified in Fig.11."

31/Figure 15. "Comparison of vertical wind velocity distributions for RPA and cloud radar for Flight 38. "Radar flight altitude" corresponds to 4 hours of cloud radar measurements at RPA flight altitude. "Radar flight time" corresponds to the cloud radar measurements during the RPA flight period at the same altitude as the RPA. The time series are defines in Fig.14. RPA measurements are divided into periods defined in Fig.13 ("cloud", "no cloud" and "broken clouds" periods). The cloud radar detected cloud only for the "broken clouds" period during the RPA flight."

34/Figure 18. "Comparison of normalized vertical wind velocity distributions for RPA during Flight 30 and cloud radar at RPA flight altitude (750 m.asl). "Radar flight altitude" period is defined in Fig.17. RPA measurements are divided into "cloud", and "no cloud" periods. The envelope of each period is plotted based on the minimum and maximum number per bin vertical velocity distributions on a leg-by-leg basis."

*14/23: 'to remove RPA motion from the wind vectors measured by the 5-hole probe' does not describe the method. It is not a data correction, but a coordinate-system transformation.*

Response: We agree; the sentence has been updated.

In the manuscript 14/23 : "Atmospheric winds in the Earth's coordinate system are derived using the equations described in Lenschow et al. (1989) with the velocity of the RPA with respect to the Earth (measured by the inertial navigation system, INS), and the velocity of the air with respect to the RPA (measured by the 5-hole probe). The attitude angles measured by the INS are used for coordinate system transformation from RPA to Earth's coordinate system."

*General stuff:*
*indices like in $\sigma_{cloud}$ shall not be italic*
*almost all Figures: axis labels etc are too small*

Response: The indices are not in italic anymore. We will work with the AMT copy editor to

insure the font size in the figures meet their standards.

*Summing up: The wind-vector calculation and the analysis of the RPA data is somewhat circuitous. Spectral and TKE comparisons to sonic measurements are not convincing. The comparison of vertical-wind distributions gained from RPA and Radar (ig. 12, 15 and 18) requires plenty of discussion and explanation (pages 11 to 13) and is somewhat puzzling. I wonder if the analysis and the results could have been explained more straight forward and thus more conclusively.*

*However, it is a measurement-technology journal, and the authors demonstrate what RPAS are good for and how remote sensing of clouds can be accompanied by quite in-expensive in situ measurements. This manuscript shows (probably for the first time) how RPAS can be used also for cloud physics.*

[revised manuscript text omitted]